# MFRM: Masked Frequency-Refined Modeling for Multivariate Time Series Anomaly Detection

## Abstract

Frequency-domain information can reveal complex characteristics such as periodicity and seasonality in time series, playing a crucial role in multivariate time series anomaly detection. Since the frequency domain features a long-tailed distribution, existing temporal reconstruction models exhibit a fundamental bias toward the information-concentrated low-frequency bands, while severely underutilizing the discriminative power of fine-grained frequency details, making the detection of complex anomalies particularly challenging. In this paper, we introduce MFRM, a novel reconstruction model that strategically leverages frequency-domain information for enhanced anomaly detection. Our key innovation lies in a learnable frequency masking module that adaptively identifies and extracts frequency components most correlated with normal behavioral patterns, enabling fine-grained frequency details utilization. Furthermore, by disrupting the original spectrum of anomalous series through its frequency masking mechanism, the MFRM exacerbates reconstruction difficulties for anomalies in the time domain and offers a novel perspective to mitigate the over-generalization issue. Extensive experiments on seven benchmark datasets demonstrate MFRM's state-of-the-art performance.

## 1 Introduction

Unsupervised learning has emerged as the predominant paradigm in multivariate time series anomaly detection (MTSAD), with reconstruction-based methods representing the most prevalent approach (Shen, 2025). These methods aim to fully learn the patterns of normal series, enabling them to accurately reconstruct normal samples but fail to reconstruct anomalies, thereby detecting anomalies through reconstruction error. Despite recent progress in this approach, two critical challenges remain.

First, traditional temporal reconstruction models exhibit a strong bias toward low-frequency bands (Piao et al., 2024), while failing to fully utilize frequency-domain information (as shown in Figure 1(a) bottom), which is crucial to detect some challenging anomalies such as *seasonal*, *shapelet*, and *trend* (Wu et al., 2025). Furthermore, previous research (Xu et al., 2024b) elucidates that frequency-domain information exhibits certain redundancy, implying that not all frequency components are equally important. However, existing methods that incorporate the frequency domain (e.g., (Wu et al., 2025)) often model the entire spectrum directly. Affected by redundant components, their frequency modeling also lacks sufficient granularity, limiting their ability to detect complex anomalies through frequency details. Therefore, it is valuable to focus on fine-grained critical frequency components within redundant frequency-domain information.

Second, most reconstruction models suffer from the over-generalization issue (Song et al., 2023; Shen, 2025). This suggests that both normal data points and anomalies are accurately reconstructed. Thus, the reconstruction error cannot distinguish anomalies from normal samples, limiting anomaly detection performance. Existing methods (e.g., (Song et al., 2023)) address this issue through the design of memory networks. However, compared to optimizing high-dimensional modeling processes, disrupting the original forms of anomalies may represent a more fundamental solution.

This paper proposes a novel reconstruction model, MFRM: Masked Frequency-Refined Modeling to address the aforementioned challenges. To fully capture fine-grained details in the frequency do-

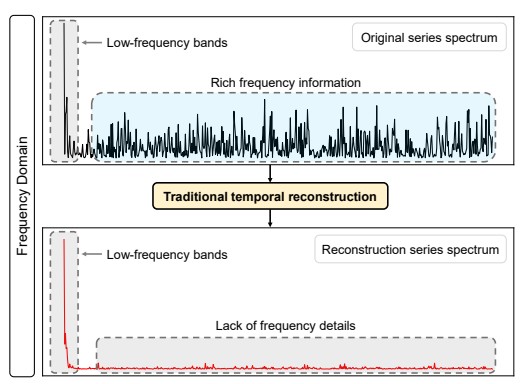 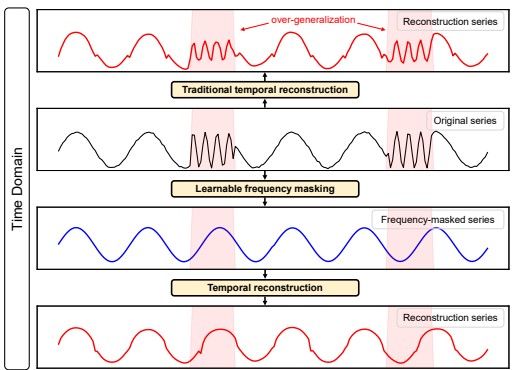

(a) Compared with the original series spectrum, the reconstruction series spectrum exhibits the bias toward low-frequency bands.

(b) Frequency masking mitigates over-generalization. Frequency-masked series represents the extracted frequencies in time domain.

Figure 1: Visualization of the two issues in reconstruction-based methods: the limited utilization of frequency-domain information and the over-generalization issue.

main, MFRM employs a learnable frequency masking module to extract specific frequencies from the series. MFRM refines normal series modeling by further focusing on these extracted specific frequencies for enhanced anomaly detection. Specifically, building upon the transformer encoder pre-trained for traditional temporal reconstruction, we employ cross-attention to leverage the extracted frequency information for guiding the model's secondary temporal reconstruction. Through training, MFRM progressively concentrates on the frequencies highly correlated with normal patterns, thereby refining the model's capacity for normal series modeling. Meanwhile, the frequency masking mechanism naturally filters out frequencies relatively irrelevant to normal patterns. Therefore, MFRM mitigates the over-generalization issue by disrupting the original spectrum of anomalous series via the frequency masking mechanism, causing anomalies converted back to the time domain to lose their original forms (as shown in Figure 1(b)) and making their reconstruction more difficult. We further employ adversarial learning to constrain attention distribution similarity between the two reconstruction stages to enhance the robustness of MFRM. Extensive experiments on seven benchmark datasets demonstrate the state-of-the-art performance of MFRM.

The contributions of this paper are as follows:

- We propose MFRM, a novel reconstruction model that strategically leverages frequency-domain information for enhanced anomaly detection. Moreover, MFRM mitigates the over-generalization issue through the filtering capability of frequency masking mechanism.

- We design a learnable frequency masking module to extract frequencies characterizing normal patterns from series. Furthermore, MFRM employs cross-attention to incorporate the information of extracted frequencies, while enhancing robustness via adversarial learning.

- We conduct extensive experiments on seven public MTSAD benchmark datasets, and the results demonstrate that MFRM outperforms state-of-the-art baselines.

## 2  RELATED WORK

### 2.1  UNSUPERVISED MTSAD

After years of development, unsupervised MTSAD has evolved into five main approaches: classification-based, density estimation-based, clustering-based, contrastive-based, and reconstruction-based methods. Classification-based methods include Isolation Forest (Liu et al., 2008) and OC-SVM (Schölkopf et al., 2001). Density estimation-based methods include LOF (Breunig et al., 2000), DAGMM (Zong et al., 2018), MMPCACD (Yairi et al., 2017), and COUTA (Xu et al., 2024a). Clustering-based methods include DeepSVDD (Ruff et al., 2018) and THOC (Shen et al., 2020). Contrastive methods include TFMAE (Fang et al., 2024) and DCdetector

(Yang et al., 2023), detecting anomalies by contrasting differences across time/frequency domains or patches/channels.

Reconstruction-based methods currently represent the most widely adopted approach in MTSAD. Early pioneering works include LSTM-VAE (Park et al., 2018), OmniAnomaly (Su et al., 2019a), GANs (Goodfellow et al., 2014) and BeatGAN (Zhou et al., 2019). Subsequently, AnomalyTrans. (Xu et al., 2022) advances the field by introducing prior associations, inspiring numerous improvements to reconstruction-based methods. Among these, MEMTO (Song et al., 2023) and H-PAD (Shen, 2025) mitigate over-generalization issue by storing normal series knowledge as retrievable memory. D3R (Wang et al., 2023) addresses data drift via dynamic decomposition, while DMamba (Chen et al., 2024) extends the Mamba model (Gu & Dao, 2023) for MTSAD.

## 2.2 FREQUENCY-DOMAIN BASED MTSAD

Frequency-domain information plays a vital role in detecting complex subsequence anomalies (Wu et al., 2025). Recent MTSAD approaches have begun incorporating frequency analysis, yet face several limitations. TFAD (Zhang et al., 2022) pioneers the integration of frequency domain and time domain, but lacks in-depth analysis of frequency-domain details. Dual-TF (Nam et al., 2024) attempts time-frequency alignment but can't resolve reconstruction-induced over-generalization. TimesNet (Wu et al., 2023) transforms time series into two-dimensional representations leveraging periodicity implicit in the frequency domain, yet this preprocessing-like approach underutilizes frequency-domain information. TFMAE (Fang et al., 2024), as a contrastive method, addresses the time series distribution shift via Top-K masking across time-frequency domains but intrinsically discards detailed high-frequency information. CATCH (Wu et al., 2025) develops a patch-based frequency-channel pipeline but still fails to address over-generalization. Existing approaches fail to focus on specific frequency information within the frequency domain and still lack sufficient attention to the over-generalization issue.

## 3 METHOD

The multivariate time series $\mathbf{X} \in \mathbb{R}^{L \times D}$ is generated by a system operating continuously over $L$ time steps, with $D$ observed variables (channels), and can be represented as a sequence of time points $\{x_1, \ldots, x_L\}$, where $x_t \in \mathbb{R}^D, t = 1, 2, ..., L$. The MTSAD problem requires the model to assign an anomaly score $\{s_1, \ldots, s_L\}$ to each time step and determine whether an anomaly occurs.

Figure 2 shows the overall architecture of the MFRM. As previously mentioned, MFRM is a reconstruction model that employs a learnable frequency masking module as a bridge to refine normal series modeling via a two-stage architecture, where both stages share the same transformer. For an input series $\mathbf{X} \in \mathbb{R}^{L \times D}$ of length $L$, the overall workflow can be formally expressed as:

$$\begin{aligned} \text{PTM Stage:} & \quad \hat{\mathbf{X}}^p = \text{Transformer}(\mathbf{X}), \\ \text{Frequency Masking:} & \quad \mathbf{X}_m = \mathcal{M}_\omega(\mathbf{X}), \\ \text{FRM Stage:} & \quad \hat{\mathbf{X}}^f = \text{Transformer}(\mathbf{X}_m, \hat{\mathbf{X}}^p), \end{aligned} \quad (1)$$

where $\mathcal{M}_\omega$ denotes the learnable frequency masking module. $\hat{\mathbf{X}}^p, \mathbf{X}_m, \hat{\mathbf{X}}^f \in \mathbb{R}^{L \times D}$. The transformer encoder employed in MFRM consists of $N$ layers. For simplicity, the following discussion will focus on one layer as an example.

## 3.1 PRIMARY TEMPORAL MODELING (PTM) STAGE

The PTM stage aims to enable the MFRM to perform traditional temporal reconstruction of the input time series in a pre-training paradigm. For the input series $\mathbf{X}$, an embedding layer is first applied:

$$\overline{\mathbf{X}} = \text{Embedding}(\mathbf{X}) = \text{Conv}_{D \to d_{\text{model}}}(\mathbf{X}) + \text{PE}, \quad (2)$$

where we employ sinusoidal positional encoding (PE), and $\overline{\mathbf{X}} \in \mathbb{R}^{L \times d_{\text{model}}}$. During the PTM stage, the Transformer employs a self-attention mechanism, denoted as $\text{Attention}(\overline{\mathbf{X}}, \overline{\mathbf{X}}, \overline{\mathbf{X}})$. This process generates the self-attention distribution $\mathcal{D}^p \in \mathbb{R}^{L \times L}$. Subsequently, residual connections are employed to reconstruct the input series through feed-forward operations and a projection head:

$$\hat{Z}^p = \text{Norm}(\text{Attention}(\overline{\mathbf{X}}, \overline{\mathbf{X}}, \overline{\mathbf{X}}) + \overline{\mathbf{X}}), \quad (3)$$

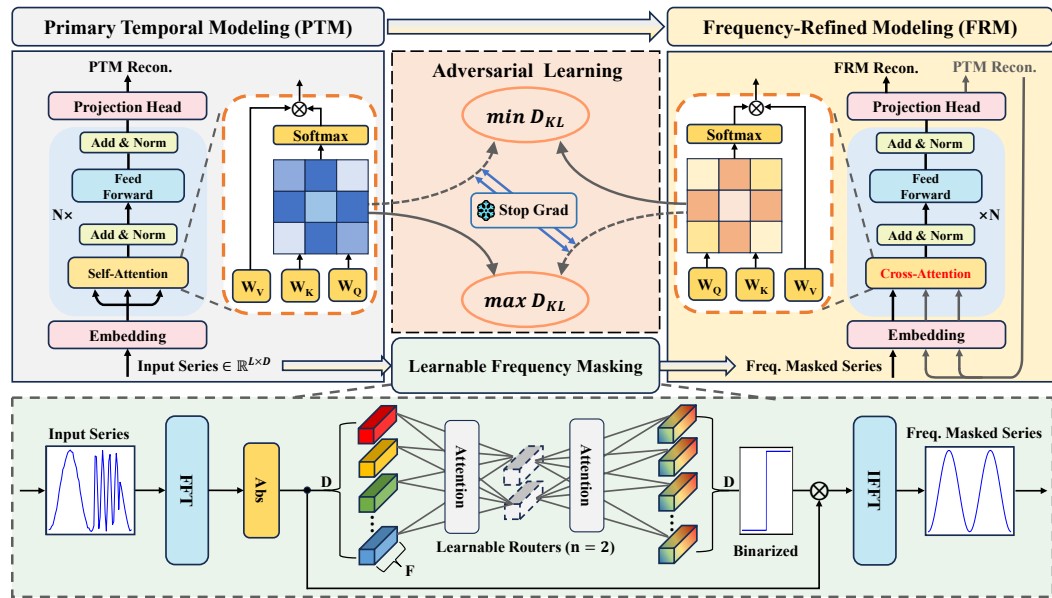

Figure 2: MFRM consists of a transformer encoder and a learnable frequency masking module. The transformer encoder only differs in the input of attention mechanism between the two reconstruction stages. In PTM stage, MFRM directly reconstructs the input series via self-attention. Then the input series generates frequency-masked series through the learnable frequency masking module. In FRM stage, it refines normal series modeling by fusing frequency-masked series and PTM's reconstruction through cross-attention. Adversarial learning aligns attention distributions between the two stages.

$$\hat{\mathbf{X}}^p = \text{Projection}(\text{Norm}(\text{Feed-Forward}(\hat{Z}^p) + \hat{Z}^p)), \tag{4}$$

where $\hat{\mathbf{X}}^p \in \mathbb{R}^{L \times D}$ is PTM's reconstruction.

## 3.2 LEARNABLE FREQUENCY MASKING MODULE

Recently, MCM's (Yin et al., 2024) learnable masking strategy has proven effective for tabular anomaly detection. Inspired by this, we design a learnable frequency masking module to selectively extract specific frequencies from the series. Specifically, we first employ the Fast Fourier Transform (FFT) to convert the input series from the time domain to the frequency domain (the specific implementation method is detailed in Appendix B.4) and obtain its magnitude spectrum:

$$\mathcal{X}_{k,:} = \sum_{t=1}^{L} x_t \cdot e^{-i(\frac{2\pi t}{L})k}, 1 \le k \le F, \quad A_{k,:} = |\mathcal{X}_{k,:}| = \sqrt{\text{Re}(\mathcal{X}_{k,:})^2 + \text{Im}(\mathcal{X}_{k,:})^2}, \tag{5}$$

where $x_t \in \mathbb{R}^D$ represents the multivariate data at time step $t$. $F$ is the number of points in the spectrum. The FFT yields $\mathcal{X} \in \mathbb{C}^{F \times D}$, with $A \in \mathbb{R}^{F \times D}$ representing its magnitude spectrum.

To generate learnable frequency masks, we design a set of learnable vectors $R$, serving as routers that represent aggregated information in the frequency-domain space. These routers interact with the magnitude spectrum $A$ through an attention mechanism to produce the mask matrix $M$:

$$M = \text{Attention}(A^T, R^T, R)^T, \tag{6}$$

where Attention denotes the attention mechanism. $R \in \mathbb{R}^{n \times F}$, where $n$ is a hyperparameter that denotes the number of routers. $M \in \mathbb{R}^{F \times D}$ represents the generated mask matrix. Subsequently, we design an activation function $f$ to binarize $M$:

$$\text{Forward:} \quad f(M_{k,d}) = \begin{cases} 1, & \text{if } [\text{Softmax}(M_{:,d})]_k \ge \frac{1}{F} \\ 0, & \text{else} \end{cases},$$

$$\text{Backward:} \quad \frac{\partial Loss}{\partial M_{k,d}} = \text{Clip}(\frac{\partial Loss}{\partial f(M_{k,d})}, -1, 1), \tag{7}$$

$$\text{Binarization:} \quad \hat{M} = f(M), \tag{8}$$

where $d \in \{1, 2, ..., D\}$, $\text{Softmax}(M_{:,d})$ operates along dimension $F$. Clip denotes the truncation operation. $\hat{M} \in \mathbb{R}^{F \times D}$ represents the binarized mask matrix.

Finally, we compute the element-wise product of $\hat{M}$ and $\mathcal{X}$ to extract specific frequencies, followed by an inverse FFT (IFFT) to transform the masked spectrum back to the time domain:

$$\hat{\mathcal{X}} = \hat{M} \odot \mathcal{X}, \quad \mathbf{X}_m = \text{IFFT}(\hat{\mathcal{X}}), \tag{9}$$

where $\hat{\mathcal{X}} \in \mathbb{C}^{F \times D}$. $\mathbf{X}_m \in \mathbb{R}^{L \times D}$ denotes the frequency-masked series in the time domain.

### 3.3 FREQUENCY-REFINED MODELING (FRM) STAGE

The FRM stage aims to refine the normal series modeling capability of MFRM by focusing on the frequency-masked series in the time domain that represent the extracted fine-grained frequencies. Specifically, $\mathbf{X}_m$ and PTM's reconstruction $\hat{\mathbf{X}}^p$ are first embedded through an embedding layer:

$$\overline{\mathbf{X}}_m = \text{Embedding}(\mathbf{X}_m), \quad \overline{\mathbf{X}}^p = \text{Embedding}(\hat{\mathbf{X}}^p). \tag{10}$$

During the FRM stage, MFRM employs a cross-attention mechanism, $\text{Attention}(\overline{\mathbf{X}}_m, \overline{\mathbf{X}}^p, \overline{\mathbf{X}}^p)$, to integrate the extracted frequency information with PTM's reconstruction. This process generates the cross-attention distribution $\mathcal{D}^f \in \mathbb{R}^{L \times L}$. Similarly, the FRM's reconstruction $\hat{\mathbf{X}}^f \in \mathbb{R}^{L \times D}$ is obtained through residual connections, feed-forward operations, and a projection head.

### 3.4 LOSS FUNCTION

The MFRM is trained end-to-end using a three-component loss function:

$$\mathcal{L} = \mathcal{L}_{rec}^p + \mathcal{L}_{rec}^f + \mathcal{L}_{ad}, \tag{11}$$

where $\mathcal{L}_{rec}^p$ and $\mathcal{L}_{rec}^f$ denote the reconstruction losses for PTM and FRM respectively, $\mathcal{L}_{ad}$ represents the adversarial loss between their attention distributions. We employ the $l_2$ norm, denoted as $|| \cdot ||_2$, to quantify the reconstruction error:

$$\mathcal{L}_{rec}^p = \sum_{i=1}^{L} \|\hat{\mathbf{X}}_i^p - \mathbf{X}_i\|_2^2, \quad \mathcal{L}_{rec}^f = \sum_{i=1}^{L} \|\hat{\mathbf{X}}_i^f - \mathbf{X}_i\|_2^2. \tag{12}$$

For adversarial learning, we adopt the method in (Fang et al., 2024), utilizing KL divergence to quantify the attention distribution similarity between the two stages:

$$\mathcal{L}_{ad} = \min_{\mathcal{D}^f} \max_{\mathcal{D}^p} \sum_{i=1}^{L} \left( D_{KL}(\mathcal{D}_i^p, \mathcal{D}_i^f) + D_{KL}(\mathcal{D}_i^f, \mathcal{D}_i^p) \right), \tag{13}$$

where $\mathcal{D}_i^p, \mathcal{D}_i^f \in \mathbb{R}^L$ denote the $i$-th row vectors of the attention distributions. The convergence of the loss function is discussed in Appendix C.2. We stop the gradients of $\mathcal{D}^p$ and $\mathcal{D}^f$ during the minimization and maximization phases, respectively, as pre-trained representations are better suited to serve as alignment targets.

### 3.5 ANOMALY SCORE

We design a hybrid anomaly score named MixScore for MFRM, which combines three loss components calculated for each individual sample through multiplicative aggregation. For timestep $t$, the $\text{MixScore}(t)$ is computed as:

$$\mathcal{S}_{rec}^p(t) = \|\hat{\mathbf{X}}_t^p - \mathbf{X}_t\|_2^2, \ \mathcal{S}_{rec}^f(t) = \|\hat{\mathbf{X}}_t^f - \mathbf{X}_t\|_2^2, \ \mathcal{S}_{ad}(t) = D_{KL}(\mathcal{D}_t^p, \mathcal{D}_t^f) + D_{KL}(\mathcal{D}_t^f, \mathcal{D}_t^p), \tag{14}$$

$$\text{MixScore}(t) = \mathcal{S}_{rec}^p(t) \cdot \mathcal{S}_{rec}^f(t) \cdot \mathcal{S}_{ad}(t). \tag{15}$$

Similar to the over-generalization mitigation perspective, the two-stage attention distributions of anomalous fragments fail to remain consistent under frequency masking, thereby contributing to anomaly detection.

# 4 EXPERIMENTS

**Datasets** We extensively evaluate MFRM on seven public benchmark datasets, including five of the most widely used public real-world benchmarks in the MTSAD domain—SMD (Su et al., 2019b), MSL (Hundman et al., 2018), SMAP (Hundman et al., 2018), SWaT (Mathur & Tippenhauer, 2016), and PSM (Abdulaal et al., 2021)—along with two additional challenging datasets featuring more complex anomalies, namely NIPS-TS-Swan (Angryk et al., 2020)and NIPS-TS-GECCO (Moritz et al., 2018). More details of the benchmark datasets are included in Appendix B.1.

**Baselines** We compare MFRM with 9 recent competitive deep learning-based unsupervised MTSAD methods, including density estimation-based COUTA (Xu et al., 2024a), contrastive learning-based TFMAE (Fang et al., 2024) and DCdetector (Yang et al., 2023), as well as reconstruction-based methods (AnomalyTrans. (Xu et al., 2022), MEMTO (Song et al., 2023), D3R (Wang et al., 2023), Dmamba (Chen et al., 2024), TimesNet (Wu et al., 2023), and CATCH (Wu et al., 2025)). Notably, TimesNet, TFMAE, and CATCH are methods that utilize frequency-domain information. For fair comparison, all methods use identical data preprocessing. We provide the key parameter configurations for each baseline method in the Appendix B.2.

Table 1: Results of label-based metrics (%) on five real-world datasets.

| Dataset | SMD | | | MSL | | | SMAP | | | SWaT | | | PSM | | |
|---|---|---|---|---|---|---|---|---|---|---|---|---|---|---|---|
| Metric | P | R | F1 | P | R | F1 | P | R | F1 | P | R | F1 | P | R | F1 |
| AnomalyTrans. | 88.88 | 91.32 | 90.08 | 90.84 | 84.43 | 92.52 | 94.53 | 93.76 | 94.14 | 90.07 | 99.77 | 94.78 | 97.22 | 94.91 | 96.05 |
| DCdetector | 86.01 | 84.52 | 85.26 | 92.10 | 92.19 | 92.14 | 93.56 | 97.94 | 95.70 | 95.31 | 97.54 | 96.45 | 97.19 | 97.00 | 97.03 |
| COUTA | 75.83 | 75.83 | 77.46 | 91.09 | 90.75 | 90.92 | 80.56 | 74.01 | 77.15 | 95.36 | 68.79 | 81.51 | **99.76** | 86.77 | 92.81 |
| MEMTO | 87.96 | 96.58 | 92.47 | 91.00 | **95.28** | 93.04 | 93.84 | **99.65** | 96.66 | 92.33 | 99.08 | 95.91 | 97.41 | 98.10 | 97.75 |
| D3R | 87.74 | 96.09 | 91.91 | 91.77 | 94.33 | 93.03 | 92.23 | 96.11 | 94.21 | 83.09 | 83.00 | 83.04 | 93.84 | **99.11** | 96.45 |
| DMamba | 92.57 | 54.04 | 68.24 | **93.69** | 64.06 | 76.09 | **95.10** | 52.98 | 68.05 | 94.11 | 86.75 | 90.28 | 98.26 | 82.89 | 89.91 |
| TimesNet | **94.53** | 82.56 | 88.14 | 90.00 | 90.12 | 90.06 | 80.99 | 69.10 | 74.58 | 90.63 | 91.37 | 90.94 | 96.79 | 96.95 | 96.82 |
| TFMAE | 90.83 | 90.27 | 90.46 | 93.01 | 95.21 | **94.33** | 95.00 | 97.97 | 96.46 | 95.10 | **100.00** | 97.49 | 97.70 | 98.24 | 97.97 |
| CATCH | 45.38 | **98.22** | 62.08 | 66.86 | 94.28 | 78.24 | 82.42 | 56.88 | 67.31 | 84.94 | 90.39 | 87.58 | 96.68 | 98.23 | 97.45 |
| MFRM | 93.38 | 96.24 | **95.05** | 93.26 | 93.18 | 93.22 | 94.91 | 99.63 | **97.21** | **97.35** | 99.23 | **98.28** | 98.37 | 98.99 | **98.68** |

Table 2: Results of score-based metrics (%) on five real-world datasets. AR and AP denote AUC-ROC and AUC-PR.

| Dataset | SMD | | MSL | | SMAP | | SWaT | | PSM | | Average | |
|---|---|---|---|---|---|---|---|---|---|---|---|---|
| Metric | AR | AP | AR | AP | AR | AP | AR | AP | AR | AP | AR | AP |
| AnomalyTrans. | 44.12 | 3.99 | 48.47 | 10.50 | 52.63 | 13.41 | 38.07 | 11.32 | 49.22 | 27.83 | 46.50 | 13.41 |
| DCdetector | 49.13 | 4.13 | 49.19 | 10.82 | **56.81** | **14.75** | 50.91 | 12.26 | 49.88 | 25.00 | 51.18 | 13.39 |
| COUTA | 49.70 | 4.14 | 50.25 | 10.55 | 49.53 | 12.77 | 49.43 | 12.13 | 51.78 | 30.29 | 50.14 | 13.98 |
| MEMTO | 50.12 | 4.86 | 49.84 | 10.57 | 50.92 | 12.95 | 77.46 | 33.72 | 50.25 | 28.15 | 55.72 | 18.05 |
| D3R | 64.20 | 12.24 | 65.26 | **16.99** | 41.35 | 10.62 | 56.65 | 13.30 | 50.03 | 26.31 | 55.50 | 15.89 |
| DMamba | 52.06 | 6.39 | 50.65 | 10.82 | 49.97 | 12.79 | 50.15 | 12.17 | 50.30 | 28.04 | 50.63 | 14.04 |
| TimesNet | 51.60 | 7.33 | 52.03 | 12.77 | 50.88 | 14.03 | 51.13 | 13.67 | 53.28 | 32.66 | 51.78 | 16.09 |
| TFMAE | 35.76 | 3.68 | 51.93 | 11.06 | 54.96 | 14.56 | 50.22 | 12.13 | 48.21 | 27.28 | 48.22 | 13.74 |
| CATCH | 78.74 | **16.35** | **66.21** | 14.53 | 53.11 | 14.37 | 39.19 | 22.66 | 65.83 | 43.96 | 60.62 | 22.37 |
| MFRM | **79.88** | 16.25 | 60.56 | 14.79 | 56.32 | 13.77 | **88.49** | **76.86** | **71.55** | **51.09** | **71.36** | **34.55** |

**Setup** To comprehensively evaluate MFRM's performance, we employ both label-based metrics—including precision (P), recall (R), and F1-score with point adjustment (PA, (Xu et al., 2018))—and score-based metrics: the Area Under the Receiver Operating Characteristic Curve (AUC-ROC) and the Area Under the Precision-Recall Curve (AUC-PR). These metrics are widely used in MTSAD. For label-based evaluation, we first apply Softmax to MixScore along the L-dimension to obtain anomaly probabilities, which are then thresholded using dataset-specific proportion parameters $r$ (0.8% for MSL/SMAP, 0.45% for SMD, 0.7% for PSM, 0.27% for SWaT, 0.25% for NIPS-TS-GECCO and 1% for NIPS-TS-Swan). In Appendix B.3, we explain the basis for specifying $r$. However, many works have demonstrated that PA can lead to faulty performance evaluations (Wang et al., 2023; Huet et al., 2022). To ensure a fair evaluation of MFRM, we employ multiple novel metrics in the Appendix C.1. Implementation details are provided in the Appendix B.4. The operational efficiency of the MFRM is discussed in the Appendix B.5.

## 4.1 MAIN RESULTS

Tables 1, 2 and 3 present the performance comparison between MFRM and 9 competitive methods on seven benchmark datasets. The results demonstrate that MFRM achieves either optimal or sub-optimal performance across all datasets in terms of both label-based metrics and score-based metrics. Compared with the state-of-the-art method CATCH, MFRM yields average improvements of 10.74% and 12.18% in AUC-ROC and AUC-PR across five real-world datasets. Meanwhile, the MFRM also demonstrates strong competitiveness on two challenging datasets containing more complex anomalies. In summary, while other methods only achieve satisfactory performance on specific datasets or a subset of evaluation metrics, extensive experiments demonstrate that the MFRM adapts well to diverse datasets within a fixed framework while delivering consistently strong performance across multiple metrics. Beyond the numerical results, visualizations of the anomaly detection performance of MFRM on the synthetic dataset are provided in the Appendix C.3. Additionally, we evaluate the performance of MFRM on a recently proposed benchmark in Appendix C.9.

Table 3: Results of multi metrics on two challenging datasets.

| Dataset | NIPS-TS-G. | | | NIPS-TS-S. | | |
|---|---|---|---|---|---|---|
| Metric | AR | AP | F1 | AR | AP | F1 |
| AnomalyTrans. | 11.86 | 0.88 | 19.20 | 41.55 | 29.63 | 73.09 |
| DCdetector | 40.30 | 0.95 | 28.35 | 42.63 | 29.89 | 73.56 |
| COUTA | 49.50 | 1.05 | 30.08 | 51.53 | 34.67 | 61.35 |
| MEMTO | 58.20 | 2.45 | 69.60 | 50.68 | 33.44 | 73.91 |
| D3R | 80.32 | 12.39 | 58.66 | 53.40 | 40.97 | 67.57 |
| DMamba | 63.23 | 14.85 | 44.23 | 51.41 | 34.44 | 74.53 |
| TimesNet | 55.33 | 10.37 | 25.38 | 54.49 | 41.31 | 65.81 |
| TFMAE | 49.29 | 1.07 | 42.87 | 48.35 | 32.18 | 73.43 |
| CATCH | 97.22 | 49.20 | 79.48 | 52.03 | 44.46 | 75.85 |
| MFRM | 87.18 | 31.67 | 80.18 | 70.77 | 61.93 | 74.79 |

## 4.2 MODEL ANALYSIS

### 4.2.1 ABLATION STUDY

As illustrated in Table 4, we conduct ablation studies on key components of the MFRM framework across five datasets. First, the FRM-only architecture (without PTM pre-training, reconstruction through self-attention) yields performance gains of 5.61% (80.44%-86.05%), demonstrating the effectiveness of the learnable frequency masking module. Second, the MFRM architecture demonstrates superior performance compared to the FRM-only approach, indicating that the two-stage method is more effective in integrating time-frequency domain details. Moreover, with respect to the similarity constraint, the adversarial learning approach outperforms the minimization strategy, as it enables the model to maintain better robustness while constraining attention consistency. Finally, for the anomaly criterion, MixScore achieves a substantial 7.49% improvement (89.00%-96.49%) compared to using reconstruction error alone. These results collectively validate the efficacy of each designed component in MFRM. We also conduct ablation studies on the anomaly criterion and transformer architecture in Appendix C.4.

Table 4: Ablation study of MFRM on F1-score (%) across five real-world benchmark datasets.

| Architecture | Criterion | Freq. Mask | Constraint | SMD | MSL | SMAP | SWaT | PSM | Average |
|---|---|---|---|---|---|---|---|---|---|
| PTM-only | Recon | × | None | 72.52 | 88.24 | 69.97 | 79.34 | 92.15 | 80.44 |
| FRM-only | Recon | ✓ | None | 86.97 | 88.83 | 70.73 | 88.13 | 95.60 | 86.05 |
| MFRM | Recon | ✓ | None | 90.59 | 92.52 | 72.13 | 88.96 | 95.97 | 88.03 |
| | Recon | ✓ | Min | 89.50 | 92.48 | 71.99 | 92.07 | 96.85 | 88.58 |
| | MixScore | ✓ | Min | 91.23 | 93.07 | 93.29 | 95.76 | 97.14 | 94.10 |
| | Recon | ✓ | Adversarial | 90.52 | 92.54 | 72.00 | 92.78 | 97.18 | 89.00 |
| | MixScore | ✓ | Adversarial | **95.05** | **93.22** | **97.21** | **98.28** | **98.68** | **96.49** |

### 4.2.2 PARAMETER SENSITIVITY

We evaluate MFRM's parameter sensitivity (Figure 3), focusing on key hyperparameters in the model architecture: the number of routers $n$, sliding window size $L$, Transformer encoder layers $N$, and $d_{model}$. As evidenced by the Figure 3, MFRM demonstrates insensitivity to both the number of routers and Transformer encoder layers. By fixing $n = 5$ and $N = 3$, MFRM achieves satisfactory comprehensive performance. For the sliding window size, we set $L = 100$ after bal-

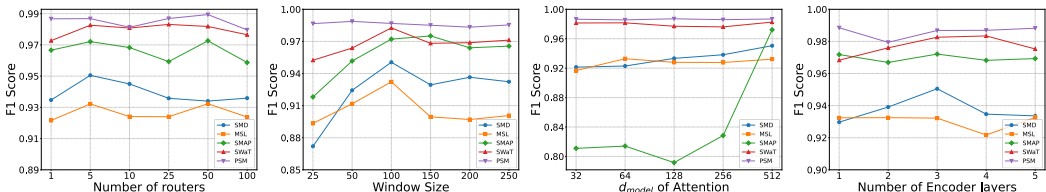

Figure 3: Parameter sensitivity studies of main hyper-parameters in MFRM.

ancing the model performance with computational efficiency. Furthermore, MFRM achieves better performance with increased $d_{\mathrm{model}}$, so we set $d_{\mathrm{model}}$ to 512 to get the best performance.

### 4.3 DISCUSSION OF FREQUENCY MASKING

#### 4.3.1 DIFFERENT FREQUENCY MASKING STRATEGIES

The learnable frequency masking module is a core component of MFRM. To validate its effectiveness, we systematically explore different masking strategies. Notably, all masking strategies are channel-independent for each time series sample.

- **Random Masking** samples frequencies with probability $p_m$ and masks others with zeros, as in (Fu & Hu, 2025).

- **Shuffle Masking** randomly samples the frequencies with probability $p_m$ and masks them with a random draw from their empirical marginal distribution, as proposed in (Bahri et al., 2022).

- **MCM Masking** employs a data-related learnable soft masking strategy that assigns different weights as masks based on input data, as proposed in (Yin et al., 2024).

- **Top-K Masking** selects frequencies based on the magnitude spectrum, retaining only the top $k\%$ as masking result, as in (Fang et al., 2024).

- **Gumble Masking** utilizes Gumbel-Softmax (Jang et al., 2017) instead of binarized activation functions, while employing an identical mask generation paradigm to our approach.

In our experiments, $p_m$ is set to 50% and $k$ to 75%. As discussed in (Fang et al., 2024), lower values of $k$ degrade the performance of the model. Table 5 shows the average performance across seven datasets under various masking strategies. Our method achieves the best performance, improving by 3.06% in AUC-ROC, 0.98% in AUC-PR, and 1.30% in F1-score over the suboptimal approach. We observe that learnable strategies generally outperform random ones. The MCM strategy, using soft masking, blends all frequency bands through linear combinations rather than precisely extracting individual frequencies, leading to inferior performance. The Top-K strategy completely ignores high-frequency band information, while the Gumbel strategy also relies on sampling and lacks precise guidance compared to the binarized activation function. In summary, our learnable masking strategy relies entirely on model learning without sampling or stochasticity, enabling precise extraction of specific frequencies through a binarized mask matrix.

Table 5: Average performance (%) comparison of different masking strategies across seven datasets.

| Strategy | Rand. | Shuffle | MCM | Top-K | Gumble. | Ours |
|---|---|---|---|---|---|---|
| AUC-ROC | 69.88 | 68.99 | 70.10 | 70.22 | 70.48 | **73.54** |
| AUC-PR | 37.03 | 36.62 | 37.07 | 37.01 | 36.98 | **38.05** |
| F1-score | 87.80 | 89.17 | 88.26 | 89.65 | 89.76 | **91.06** |

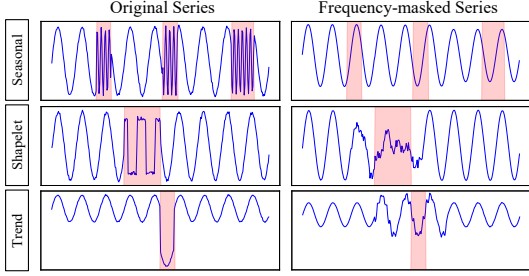

Figure 4: Frequency filtering mitigates over-generalization issue.

### 4.3.2 VALIDITY OF LEARNABLE FREQUENCY MASKING

The learnable frequency masking module serves a dual purpose: extracting specific frequencies that are highly correlated with normal patterns in the series while filtering out other frequencies. This section will analyze the effectiveness of the learnable frequency masking from these two perspectives.

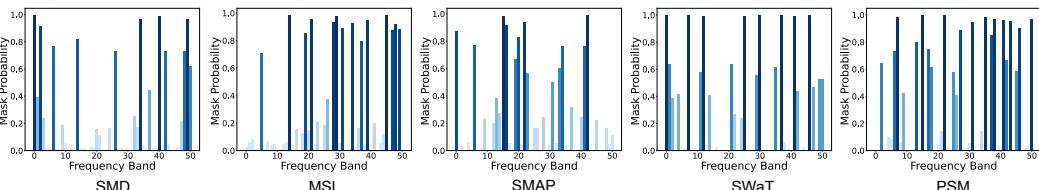

Figure 5: The average probability distribution of extracted frequencies across five datasets though the learnable frequency masking module. A higher value indicates that the frequency is extracted more frequently.

**Frequency Filtering Mitigates Over-generalization** Firstly, since the learnable frequency masking is optimized on normal series, it tends to select frequencies corresponding to normal behavior while filtering out "unseen" frequencies deviating from normal patterns. Some subsequence anomalies that are prone to cause over-generalization often exhibit spectra distinct from normal data. Thus, by applying the frequency masking, anomalous frequencies can be filtered out while normal ones are preserved, thereby fundamentally mitigating the over-generalization issue. For intuitive understanding, Figure 4 visualizes the frequency-masked series of three subsequence anomalies (as defined in (Lai et al., 2021)) in the time domain. It is evident that for different types of anomalies, thanks to the frequency filtering capability of the learnable frequency masking mechanism, they lose the original anomalous forms in the time domain after frequency masking, thereby making their reconstruction more difficult and thus mitigating the over-generalization issue. Appendix C.5 provides the corresponding frequency-domain visualization.

**Ability of Extracting Specific Frequencies** Secondly, it is equally important to understand how the module selects specific frequencies to achieve fine-grained utilization of frequency-domain information. Figure 5 visualizes the average probability distribution of the extracted frequencies across five datasets, computed by averaging across data points and channels. The results demonstrate that: (1) Our method exhibits extraction capability across all frequency bands, distinctly differing from the Top-K strategy and showcasing the advantage of a learnable approach; (2) The extraction probabilities vary significantly across different frequencies in all five datasets, with peak probabilities concentrated in specific frequencies. This indicates MFRM's ability to focus on specific frequencies within the series, markedly contrasting with a random strategy (which would exhibit a uniform distribution under the law of large numbers). Appendix C.6 provides further analysis demonstrating that the frequencies extracted by the learnable frequency masking module are indeed highly correlated with normal patterns. Furthermore, Appendix C.7 presents a case study designed to illustrate how the learnable frequency mask leverages frequency extraction and filtration mechanisms to detect complex anomalies caused by spectral discrepancies. Appendix C.8 presents additional visualizations of the frequency masking.

## 5 CONCLUSION

In this paper, we present a novel MTSAD method named MFRM. Specifically, we design a learnable frequency masking module to extract specific frequencies that characterize normal patterns from the series, and further guide MFRM in focusing on these fine-grained frequency details for enhanced anomaly detection. Essentially, MFRM addresses two challenges faced by temporal reconstruction models in MTSAD through the learnable frequency masking strategy: (1) the insufficient fine-grained utilization of frequency-domain information, and (2) the over-generalization issue. Comprehensive experiments on seven benchmark datasets demonstrate MFRM's superior performance over 9 recent competitive baselines. Future work will extend MFRM to broader time series applications (e.g., forecasting and classification).

## 6 REPRODUCIBILITY STATEMENT

To ensure the reproducibility of our work, we provide detailed implementation specifics of the MFRM (including dataset processing procedures) in Appendix B.4 along with an anonymous link to the source code, while key parameter configurations for all baseline methods are discussed in Appendix B.2.

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

## A  THE USE OF LARGE LANGUAGE MODELS (LLMS)

In the process of writing this paper, we employ OpenAI's GPT-4o to polish and refine our writing. Specifically, we use it to assist with grammar checking, improving sentence fluency, and enhancing the overall clarity of our pre-written text. All intellectual ideas, data interpretations, and conclusions are solely our own. We have thoroughly reviewed and edited all output from the model and take full responsibility for the content of this published work.

## B  EXPERIMENTAL DETAILS

### B.1  DATASETS

Table 6 lists statistics of the 7 public benchmark datasets.

Table 6: Statistics of the seven benchmarks, including the domain of dataset, the number of feature dimensions, as well as the total length of the training, valid and test sets. AR denotes the abnormal proportion of the test set.

| Dataset | Domain | Train | Valid | Test | Dimension | AR |
|---|---|---|---|---|---|---|
| SMD | Server Machine | 566724 | 141681 | 708420 | 38 | 4.2% |
| MSL | Spacecraft | 46653 | 11664 | 73729 | 55 | 10.5% |
| SMAP | Spacecraft | 108146 | 27037 | 427617 | 25 | 12.8% |
| SWaT | Water Treatment | 396000 | 99000 | 449919 | 51 | 12.1% |
| PSM | Server Machine | 105984 | 26497 | 87841 | 25 | 27.8% |
| NIPS-TS-GECCO | Water Treatment | 55408 | 13852 | 69261 | 9 | 1.1% |
| NIPS-TS-Swan | Machinery | 48000 | 12000 | 60000 | 38 | 32.6% |

### B.2  BASELINE HYPERPARAMETER SETTINGS

For each baseline method, we strictly adhere to the hyperparameter configurations recommended in their original papers. Furthermore, we conduct extensive hyperparameter searches across multiple sets and select the optimal configurations based on these evaluations to ensure a comprehensive and fair assessment of each method's performance. We employ a **non-overlapping sliding window** approach to partition the dataset for all methods, thereby reducing the computational costs associated with model training and inference. The hyperparameters for the baseline methods are set as follows:

- **AnomalyTransformer**: The channel number of hidden states $d_{model}$ is 512, and the number of heads is 8, the number of layers is 3. The loss function hyperparameter $k$ is set to 3. The window size is set to 100. The anomaly ratio hyperparameter $r$ for each dataset is set consistent with its respective open-source implementation.

- **DCdetactor**: The channel number of hidden states $d_{model}$ is 256, and the number of heads is 4, the number of layers is 3. The loss function hyperparameter $k$ is set to 3. The anomaly ratio, patch size and window size hyperparameters for each dataset are set consistent with its respective open-source implementation.

- **COUTA**: The channel number of hidden states is set to 128, and the kernel size is 2. The window size is set to 100.

- **MEMTO**: The number of memory is 10, the channel number of hidden states $d_{model}$ is 512. The loss function hyperparameter $\lambda$ is set to 0.01. The window size is set to 100. The anomaly ratio for each dataset is set consistent with its respective open-source implementation.

- **D3R**: The number of blocks is set to 2, and the dimension of hidden layer is set to 512, the dimension of FCN is set to 2048, the number of heads is 8. The anomaly ratio is set from 0.005 and 0.01. The window size is set to 64.

- **DMamba**: The number of blocks is set from 1,3,5,10 and the channel number of hidden states $d_{model}$ is 256. The window size is set to 100.

- **TimesNet**: The hyperparameter Top-$k$ is set to 5, the channel number of hidden states $d_{model}$ is 256. The number of layers is 3, and the number of kernels is 6. The window size is set to 100.

- **TFMAE**: The frequency masking ratio is set from 0.1, 0.2, 0.3, 0.4. The time masking ratio is set from 0.05, 0.25, 0.55, 0.65. The channel number of hidden states $d_{model}$ is 128 and the number of layers is 3. The window size is set to 100. The anomaly ratio for each dataset is set consistent with its respective open-source implementation.

- **CATCH**: The number of layers is set from 1, 2, 3. The channel number of hidden states $d_{model}$ is 128, the number of heads is set from 4, 8, 12, 16. The window size is set from 96, 192. The patch size, the loss function hyperparameter $\lambda_1$, $\lambda_2$ and $\lambda_3$, the hyperparameter of anomaly score $\lambda_{score}$ and the anomaly ratio for each dataset are set consistent with its respective open-source implementation.

### B.3 THRESHOLD SELECTION

As for the threshold selection strategy, we adopt the same **Gap Statistic method (Tibshirani et al., 2002) in K-Means** as used in AnomalyTransformer (Xu et al., 2022). Specifically, we analyze the distribution of anomaly scores on the unlabeled validation set to ensure that the threshold could effectively identify clusters with notably high anomaly score. Based on Table 7, we determine the hyperparameter $r$ for each dataset (0.8% for MSL/SMAP, 0.45% for SMD, 0.7% for PSM, 0.27% for SWaT, 0.25% for NIPS-TS-GECCO and 1% for NIPS-TS-Swan).

Furthermore, as observed in Table 7, directly applying a fixed threshold of 0.9 can also effectively distinguish anomaly clusters. Therefore, Table 8 compares the F1-scores of two methods—hyperparameter-based partitioning and fixed-threshold partitioning—across seven datasets. The results demonstrate that a fixed threshold can also achieve comparable performance, offering practical value for real-world anomaly detection applications.

Table 7: Statistical results of anomaly score distribution on the validation set. We count the number of time points with corresponding values in several intervals.

| Anomaly Score Interval | SMD | MSL | SMAP | SWaT | PSM | GECCO | Swan |
|:---:|:---:|:---:|:---:|:---:|:---:|:---:|:---:|
| $(0, +\infty]$ | 141681 | 11664 | 27037 | 99000 | 26497 | 13852 | 12000 |
| $(0, 0.1]$ | 140734 | 11532 | 26789 | 98673 | 26270 | 13814 | 11841 |
| $(0.1, 0.9]$ | 291 | 37 | 27 | 60 | 44 | 5 | 31 |
| $(0.9, +\infty]$ | 656 | 95 | 221 | 267 | 183 | 33 | 128 |
| Ratio of $(0.9, +\infty]$ | 0.46% | 0.81% | 0.82% | 0.27% | 0.69% | 0.24% | 1.07% |

Table 8: F1-score (%) performance comparison of the two threshold selection methods.

| Dataset | SMD | MSL | SMAP | SWaT | PSM | NIPS-TS-G. | NIPS-TS-S. |
|:---:|:---:|:---:|:---:|:---:|:---:|:---:|:---:|
| Chosen by $r$ | **95.05** | **93.22** | **97.21** | **98.28** | **98.68** | 80.18 | **74.79** |
| Fixed threshold | 93.29 | 93.19 | 94.59 | 97.81 | 98.31 | **80.90** | 74.55 |

### B.4 IMPLEMENTATION DETAILS

In the implementation, we employ a non-overlapping sliding window of size $L = 100$ and Standard-Scaler for preprocessing across all datasets. For evaluation, MFRM retains identical hyperparameters in its architecture: 3 Transformer encoder layers with $d_{model} = 512$ and 8 attention heads. The

number of routers in the frequency masking module is fixed at 5, and we fix the batch size at 256. All experiments are conducted on an Ubuntu 18.04 system using PyTorch 1.12.1 with two Tesla T4 16GB GPUs, with a random seed of 42. We repeat each experiment three times and report the mean results. We employ the Adam optimizer, with an initial learning rate of $10^{-2}$ for the SMAP dataset and $10^{-4}$ for all other datasets. The learning rate decays by a factor of 0.95 after each epoch. The training process is early stopped within 10 epochs. For time-frequency domain transformation, we employ the $rfft$ (real-valued Fast Fourier Transform) and $irfft$ (its inverse) to eliminate the effects of conjugate frequencies. So that the value of $F$ is typically $L/2 + 1$. The code of MFRM is available at `https://anonymous.4open.science/r/MFRM-7F22/`.

### B.5 DISCUSSION OF OPERATIONAL EFFICIENCY

Model efficiency analysis is becoming increasingly critical for real-world deployment. In Table 9, we present a quantitative evaluation of the per-epoch runtime overhead, the number of parameters (NParams), and maximum memory usage of MFRM, specifically considering the following three aspects: (i) different model architectures, (ii) whether an additional learnable frequency masking module is adopted, and (iii) varying model complexities ($d_{model}$).

**Different Architectures** We compare the efficiency of three model architectures: PTM, MFRM, and encoder–decoder (utilizing the encoder during PTM while employing the decoder in FRM). Employing an encoder-only architecture, MFRM adopts a two-stage training strategy. Although it incurs higher computational costs compared to the single-stage PTM, it generally outperforms the encoder–decoder architecture while reducing the number of parameters by nearly half, thereby significantly cutting deployment costs.

**Additional Module** The learnable frequency masking is an additional module introduced in MFRM. Its effectiveness has been validated in Table 4, while the efficiency analysis provided in Table 9 demonstrates that this module only incurs a minimal computational overhead, yet contributes to a significant performance improvement in the model.

**Varying Model Complexities** MFRM is designed based on the Transformer architecture, and its complexity is primarily determined by $d_{model}$. The results summarized in Table 9 further indicate that as $d_{model}$ increases, the computational and deployment costs of MFRM increase significantly, leading to reduced efficiency.

Table 9: Efficiency analysis of MFRM.

| Efficiency | | Epoch Time (s) | NParams (M) | Max Memory (GB) |
|---|---|---|---|---|
| Architecture | PTM | 3.97 | 4.86 | 3.66 |
| | MFRM | 7.06 | 4.92 | 5.64 |
| | encoder-decoder | 7.86 | 9.68 | 5.81 |
| Freq. Masking | w/o Freq. Masking | 6.93 | 4.86 | 5.60 |
| | w/ Freq. Masking | 7.06 | 4.92 | 5.64 |
| $d_{model}$ | 128 | 2.32 | 0.39 | 3.00 |
| | 256 | 3.40 | 1.31 | 3.87 |
| | 512 | 7.06 | 4.92 | 5.64 |
| | 1024 | 19.62 | 19.22 | 9.41 |

## C ADDITIONAL EXPERIMENTS

### C.1 MULTI METRICS RESULTS

In addition to the three evaluation metrics employed in the main text, we employ a more comprehensive set of metrics (as shown in Table 10) to rigorously assess the performance of MFRM, with comparative analysis against the state-of-the-art method, CATCH. Specifically, Acc denotes the accuracy. Affiliation precision (Aff-P) and recall (Aff-R) quantify alignment accuracy by measuring spatial discrepancies between predicted events and ground truth annotations (Huet et al.,

2022). Range-based evaluation metrics include R-AR (Range-AUC-ROC) and R-AP (Range-AUC-PR), which are derived through label transformation techniques (Paparrizos et al., 2022). Furthermore, Volume Under the Surface (VUS) metrics have recently garnered significant attention from researchers, including V-ROC (VUS-ROC) and V-PR (VUS-PR) (Paparrizos et al., 2022). A comparative analysis reveals that the MFRM maintains strong competitiveness across multiple evaluation metrics, demonstrating its exceptional robustness and comprehensive performance.

Table 10: A comprehensive evaluation of MFRM versus CATCH using additional metrics (%) across seven benchmark datasets.

| Dataset | Method | Acc | F1 | Aff-P | Aff-R | R-AR | R-AP | V-ROC | V-PR |
|---|---|---|---|---|---|---|---|---|---|
| SMD | MFRM | **99.46** | **95.05** | 67.86 | **94.26** | **85.22** | 22.13 | **84.44** | 21.64 |
| | CATCH | 95.01 | 62.08 | **76.14** | 93.07 | 84.04 | **24.08** | 83.34 | **23.30** |
| MSL | MFRM | **98.57** | **93.22** | 53.73 | 95.05 | 66.71 | **23.11** | 66.02 | **22.75** |
| | CATCH | 94.48 | 78.24 | **57.38** | **96.02** | **68.56** | 20.90 | **67.81** | 20.56 |
| SMAP | MFRM | **99.27** | **97.21** | **50.01** | **97.00** | **58.48** | **15.14** | **58.38** | **15.13** |
| | CATCH | 92.93 | 67.31 | 45.42 | 70.02 | 52.55 | 12.34 | 52.50 | 12.33 |
| SWaT | MFRM | **99.57** | **98.26** | **57.63** | **97.15** | **84.13** | **63.88** | **84.03** | **64.00** |
| | CATCH | 96.89 | 87.58 | 57.15 | 84.84 | 48.12 | 27.33 | 48.23 | 27.55 |
| PSM | MFRM | **99.27** | **98.68** | 57.98 | 80.90 | **75.29** | **55.23** | **74.13** | **54.56** |
| | CATCH | 98.57 | 97.45 | **72.96** | **93.68** | 69.15 | 50.32 | 68.23 | 49.29 |
| NIPS-TS-G. | MFRM | **99.56** | **80.18** | 69.69 | 83.91 | 79.77 | 38.37 | 79.45 | 38.87 |
| | CATCH | 99.46 | 79.48 | **85.49** | **99.69** | **98.28** | **55.76** | **97.92** | **53.71** |
| NIPS-TS-S. | MFRM | 86.74 | 74.79 | 79.68 | 4.66 | **94.04** | **94.00** | **92.70** | **92.52** |
| | CATCH | **87.25** | **75.85** | **93.14** | **8.04** | 92.62 | 92.16 | 91.01 | 90.25 |
| Average | MFRM | **97.49** | **91.06** | 62.37 | **78.99** | **77.66** | **44.55** | **77.02** | **44.21** |
| | CATCH | 94.94 | 78.28 | **69.67** | 77.91 | 73.33 | 40.41 | 72.72 | 39.57 |

## C.2 CONVERGENCE ANALYSIS OF THE LOSS FUNCTION

The MFRM employs adversarial learning to constrain the similarity between the attention distributions generated in both the PTM and the FRM stages, while integrating the reconstruction loss from both phases for end-to-end training. Figures 6 and 7 illustrate the convergence trends of (1) the mean reconstruction errors and (2) the KL divergence between the two attention distributions across five benchmark datasets during training.

Overall, the reconstruction loss exhibits relatively stable convergence, eventually stabilizing near a consistent value. In contrast, the KL divergence of the attention distributions—due to the adversarial learning mechanism—does not follow a direct convergence pattern. Across the five datasets, the KL divergence loss initially increases before eventually converging. This phenomenon demonstrates that through adversarial learning during training, the MFRM dynamically prioritizes specific frequency information and enables the model to maintain superior robustness.

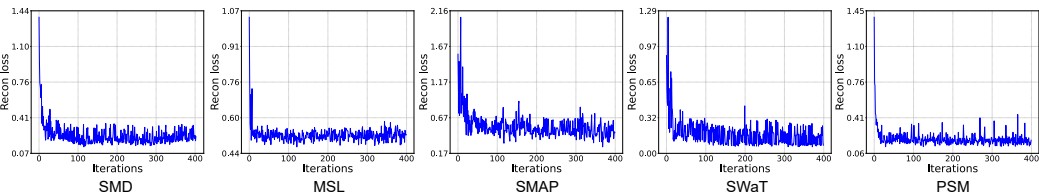

Figure 6: The convergence of average reconstruction error in both PTM and FRM stages during training.

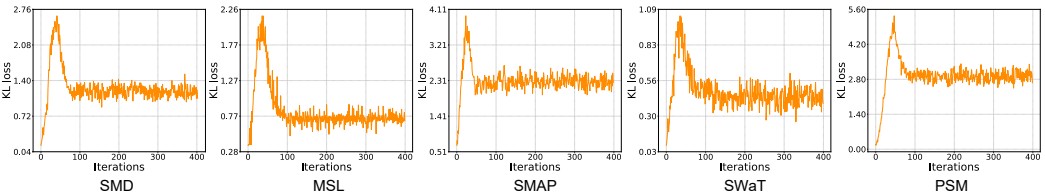

Figure 7: The convergence of KL divergence in two-stage attention distribution during training.

## C.3 ANOMALY CRITERION VISUALIZATION

To validate the effectiveness of MFRM in detecting various types of anomalies, we visualize the anomaly criterion (MixScore) of MFRM on five synthetic anomalies (Lai et al., 2021) and compare its detection performance with TFMAE, as shown in Figure 8. The results demonstrate that our MixScore can significantly suppress false detections while successfully identifying diverse anomaly types, whereas TFMAE exhibits more volatile anomaly scores that are prone to false detections. In fact, the MFRM focuses on fine-grained frequency components that are highly correlated with the normal patterns while filtering out redundant frequencies, thereby rendering the anomaly score distribution both precise and smooth. However, TFMAE lacks focused attention on specific frequency components, limiting its ability to detect complex anomalies due to interference from redundant frequencies.

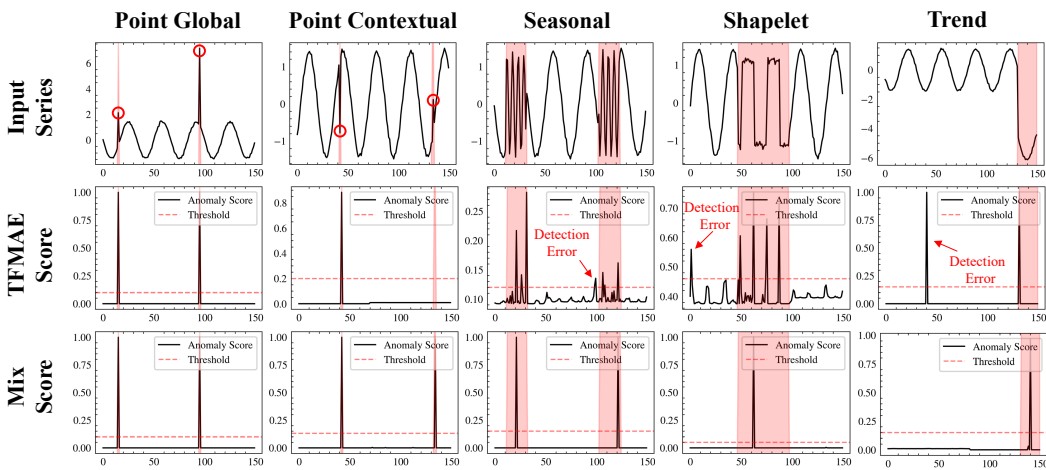

Figure 8: The anomaly detection performance of MFRM and TFMAE is comparatively visualized across five categories of synthetic anomalies, comprising two point anomaly types (global and contextual) and three subsequence anomalies (seasonal, shapelet and trend). MFRM yields nearly noise-free anomaly detection scores, whereas TFMAE exhibits significantly more volatile score fluctuations, leading to both false positives and missed detections.

## C.4 SUPPLEMENTARY ABLATION STUDIES

In addition to the main text's ablation studies on MFRM's key components, we further evaluate the anomaly criterion (Table 11) and transformer-based architecture (Table 12). Considering both performance and efficiency, our chosen product-based anomaly criterion and encoder-only architecture achieve better results.

## C.5 FREQUENCY-DOMAIN VISUALIZATION OF FREQUENCY MASKING

Figure 9 visually compares the spectral differences of three types of subsequence anomalies before and after frequency masking. When processing anomalous series, the frequency masking mechanism disrupts their original spectrum while selectively retaining only frequency components that

Table 11: Ablation study of anomaly criterion on F1-score (%) across five public benchmarks. *Combine* refers to the methodology (add or multiply) employed to integrate the three component scores.

| Combine | Criterion | | | SMD | MSL | SMAP | SWaT | PSM |
|---------|-----------|-----------|----------|------|------|------|------|------|
| | $S_{rec}^p$ | $S_{rec}^f$ | $S_{ad}$ | | | | | |
| None | ✓ | × | × | 85.82 | 88.84 | 71.64 | 87.49 | 95.00 |
| None | × | ✓ | × | 85.69 | 88.86 | 71.63 | 87.59 | 97.18 |
| None | × | × | ✓ | 90.05 | 89.64 | 92.99 | 96.02 | 98.17 |
| Multi | ✓ | ✓ | × | 90.52 | 92.54 | 72.00 | 92.78 | 97.18 |
| Add | ✓ | ✓ | ✓ | 89.74 | **93.27** | 93.32 | 96.01 | 97.99 |
| Multi | ✓ | ✓ | ✓ | **95.05** | 93.22 | **97.21** | **98.28** | **98.68** |

Table 12: Ablation study of the transformer-based architecture on AUC-ROC (AR), AUC-PR (AP), F1-score across five public benchmarks (%).

| Architecture | SMD | | | MSL | | | SMAP | | | SWaT | | | PSM | | |
|--------------|-----|-----|-----|-----|-----|-----|------|-----|-----|------|-----|-----|-----|-----|-----|
| | AR | AP | F1 | AR | AP | F1 | AR | AP | F1 | AR | AP | F1 | AR | AP | F1 |
| encoder-decoder | 78.66 | 15.85 | 93.45 | 59.59 | 14.53 | 92.44 | 56.17 | 13.45 | 96.91 | 87.95 | 76.62 | 96.98 | **71.73** | **51.19** | **98.93** |
| Ours (encoder only) | **79.88** | **16.25** | **95.05** | **60.56** | **14.79** | **93.22** | **56.32** | **13.77** | **97.21** | **88.49** | **76.86** | **98.28** | 71.55 | 51.09 | 98.68 |

characterize normal patterns, thereby filtering out other frequencies irrelevant to normal patterns (which may potentially carry significant indications of anomalies). This result provides a frequency-domain explanation for how MFRM mitigates the over-generalization issue through the learnable frequency masking.

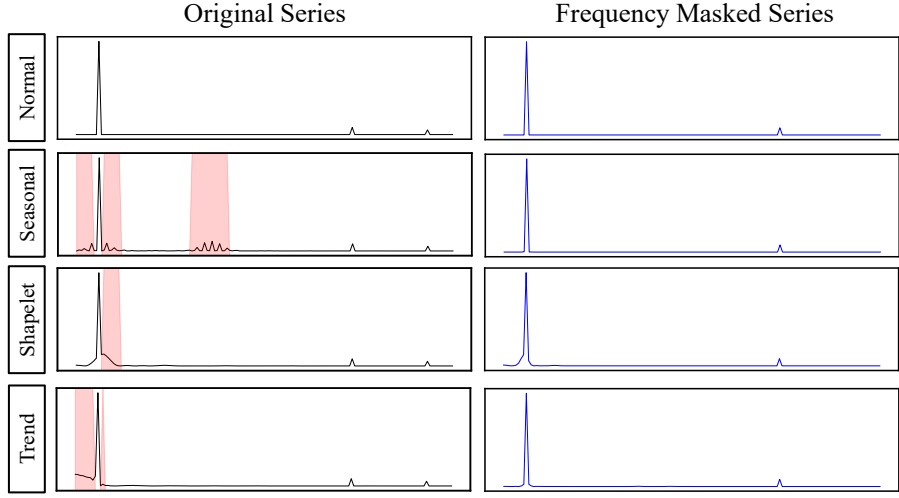

Figure 9: Frequency-domain visualization of three subsequence anomalies before and after frequency masking. The highlighted portions represent the frequency bands suppressed by frequency masking in the spectrum of the abnormal series.

## C.6 DISCUSSION OF THE EXTRACTED FREQUENCY INFORMATION

We primarily conduct the following two experiments to demonstrate that the frequencies extracted by the learnable frequency masking are highly correlated with the normal patterns.

**t-SNE Visualization** Figure 10 compares t-SNE embeddings of Top-K and our learnable masking strategies on the SMAP dataset. The learnable frequency masking (Figure 10, right) compresses normal samples into a compact cluster (blue → green), characterizing the frequency patterns common to normal series. After applying the frequency masking, anomaly samples are also incorporated into

this dense representation (orange → purple). This demonstrates that the learnable frequency masking can effectively extract frequency components that are highly correlated with normal patterns. In contrast, Top-K masking (Figure 10, left) only preserves low frequencies and yields distributions similar to the original data, underscoring its limited discriminative ability and highlighting the superiority of our learnable approach.

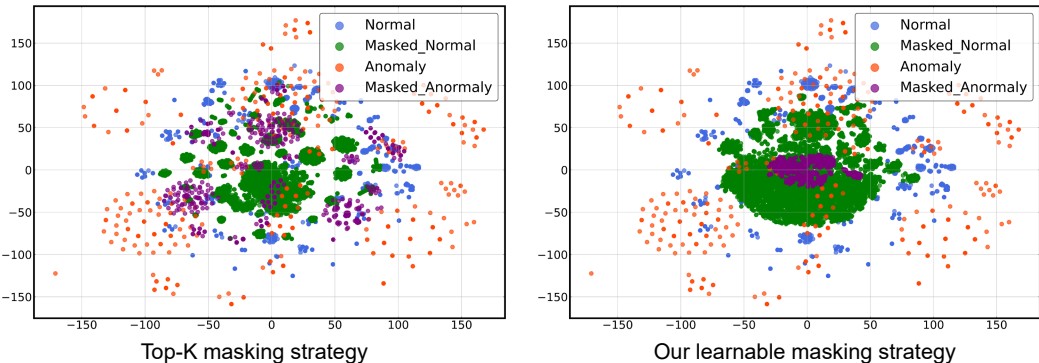

Figure 10: Comparative t-SNE visualization results between the Top-K masking strategy and our learnable approach on the SMAP dataset. The t-SNE visualization uses blue and orange to represent the original normal and abnormal samples, respectively, while green and purple denote their corresponding distributions after frequency masking.

**Reconstruction Error Analysis** We employ the pre-trained MFRM to reconstruct original series using three distinct types of information: (1) original series, (2) extracted frequency information, and (3) filtered frequency information. We present the corresponding reconstruction errors for all seven datasets in Table 13. The results demonstrate that, across all datasets, reconstructing the original series using the extracted frequency information consistently yields a lower error compared to using the filtered components and performs nearly as well as a direct reconstruction from the original series. This indicates that the extracted frequencies are highly correlated with normal patterns, whereas the filtered components are relatively irrelevant.

Table 13: Mean reconstruction errors when using three distinct types of information to reconstruct the original series, evaluated across 7 datasets.

| Recon. based | Mean Reconstruction Error | | | | | | |
|---|---|---|---|---|---|---|---|
| | SMD | MSL | SMAP | SWaT | PSM | GECCO | Swan |
| Original Series | **0.2865** | 0.5836 | **3.5645** | 0.2553 | 0.1811 | **0.6408** | 1.0331 |
| Extracted Freq. | 0.2888 | **0.5766** | 3.8654 | **0.2447** | **0.1801** | 0.7040 | **1.0033** |
| Filtered Freq. | 0.3326 | 0.5808 | 3.8695 | 0.3152 | 0.2142 | 0.7269 | 1.0117 |

## C.7 DETECTING ANOMALIES INDUCED BY SPECTRAL DISCREPANCIES

To further validate the capability of MFRM in detecting complex anomalies induced by spectral discrepancies, we design a case study. Specifically, we fix a reference spectrum (as shown in the upper left of Figure 11) and convert it back to the time domain to generate normal series for training. Consequently, we create the anomalous spectrum (as illustrated in the upper right of Figure 11) and the corresponding anomalous series by injecting varying levels of noise across the entire frequency band of the normal spectrum. The lower portion of Figure 11 displays the masked normal/anomalous spectrum after applying the learnable frequency masking.

During training on normal series, the learnable frequency masking module acquires the capability to extract frequencies highly correlated with normal patterns (as shown in the two left subfigures). While the spectrum of anomalous series exhibit deviations from normal spectrum across all frequency bands, the frequency masking operation effectively filters out prominent anomalous components while preserving relatively normal frequencies (as shown in the two right subfigures).

Therefore, MFRM can effectively disrupt the original spectrum of anomalous series through its frequency masking mechanism, preventing their successful reconstruction in the time domain, and thereby enabling the identification of complex anomalies arising from spectral discrepancies.

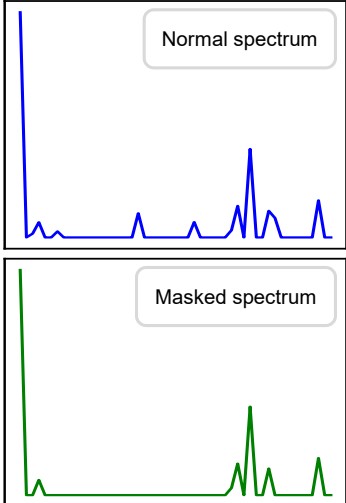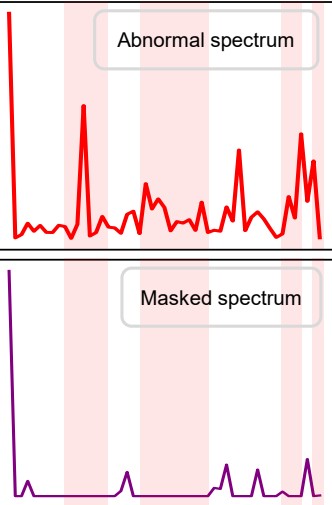

Figure 11: The spectral visualization of the case study. Highlighted regions indicate significant anomalous frequency bands.

### C.8    MORE VISUALIZATIONS ON THE EXTRACTION OF SPECIFIC FREQUENCY INFORMATION

Here, we provide more visualizations of spectrum comparisons as shown in Figure 12. It can be observed that the original series exhibit a spectrum rich in frequency-domain information, whereas the traditional temporal reconstruction series exhibit a fundamental bias toward the low-frequency bands. Our proposed learnable frequency masking enables the extraction of specific frequency components from the original spectrum, allowing the model to acquire fine-grained frequency-domain details.

### C.9    TSB-AD BENCHMARK

TSB-AD (Liu & Paparrizos, 2024) is a recently proposed benchmark for MTSAD. It provides a comprehensive discussion of three major limitations in existing MTSAD benchmark datasets—mislabeling issues, dataset bias, and feasibility for anomaly detection—and introduces two refined, high-quality time series datasets that include both univariate (TSB-AD-U) and multivariate (TSB-AD-M) versions.

Here, we select the TSB-AD-U dataset to re-evaluate the performance of MFRM and various baselines. The TSB-AD-U dataset comprises a total of 870 sub-datasets, of which the first 100 sub-datasets are chosen for our experiment. The lengths of these sub-datasets range from 1,000 to 18,397. The split between training and test sets follows the official configuration. As shown in Table 14, our MFRM maintains superior performance under the new benchmark and across multiple evaluation metrics.

### C.10    COMPARE WITH CLASSICAL STATISTICAL LEARNING METHODS

Given the potential of classical statistical learning methods to outperform deep learning approaches on certain metrics or datasets, we supplement the performance comparisons of Isolation Forest, LOF, and OC-SVM with deep learning methods across seven datasets in Tables 15, 16, and 17. The results demonstrate that these methods indeed achieve superior AUC-ROC and AUC-PR values on datasets such as SMD and PSM, surpassing most deep learning approaches. Therefore, it remains necessary to incorporate these classical methods into consideration.

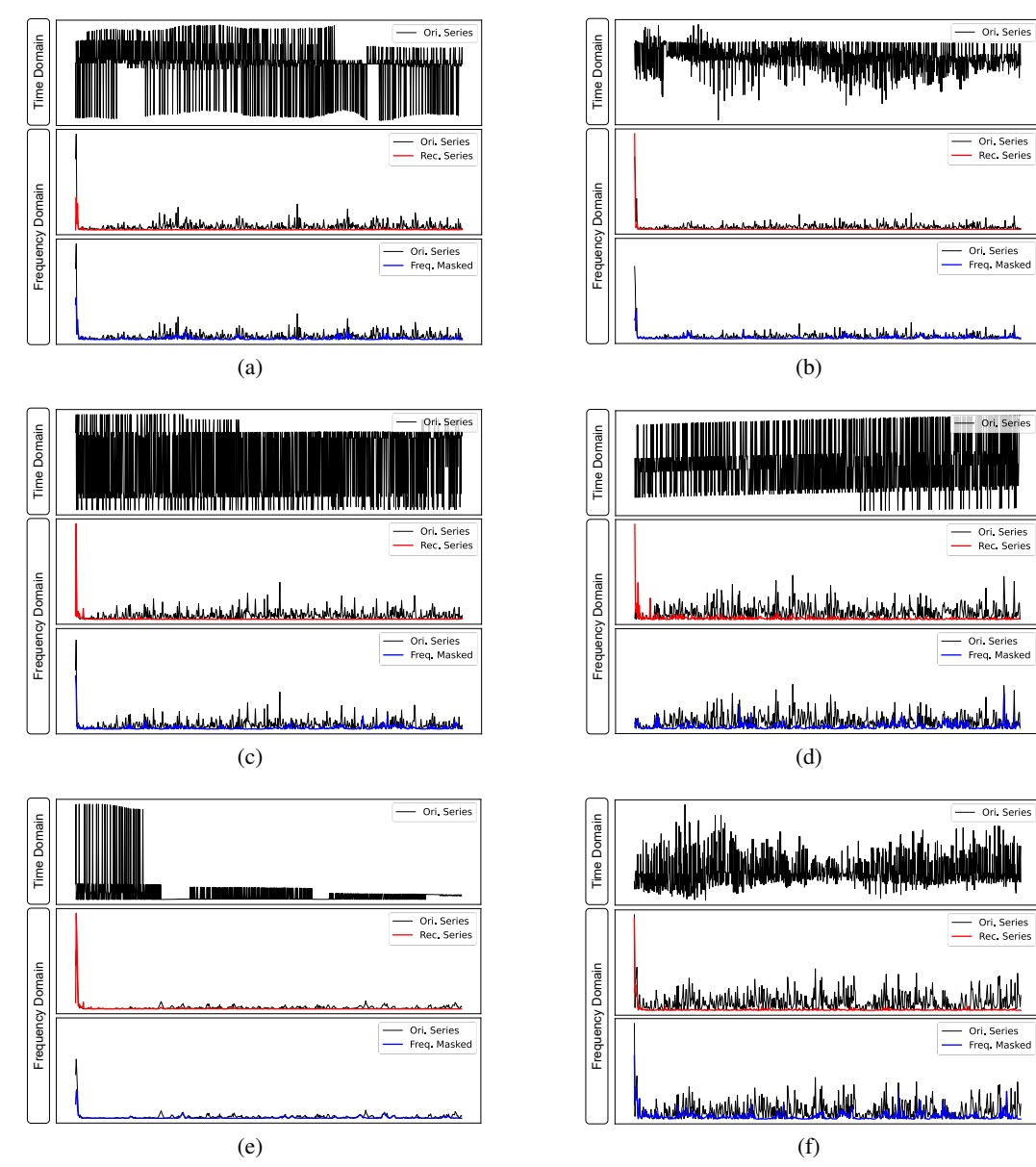

Figure 12: More visualizations of spectrum comparisons. We respectively present the spectrum of the original series, the spectrum of the reconstruction series through traditional temporal modeling, and the spectrum after masking by the learnable frequency masking module.

## C.11 MORE VISUALIZATION OF OVER-GENERALIZATION ISSUE

To verify that the over-generalization issue is indeed prevalent in reconstruction-based methods, particularly in SOTA approaches, we visualize in Figure 13 the reconstruction results of Anomaly-Transformer, MEMTO, TimesNet, CATCH, and our proposed MFRM on three types of synthetic anomalies. The results reveal that all four SOTA reconstruction-based methods exhibit varying degrees of over-generalization, especially in the case of *seasonal* and *shapelet* anomalies. In contrast, MFRM employs frequency masking to filter out anomalous frequencies unrelated to normal patterns, thereby preventing anomalies from being well reconstructed and offering a fundamental solution to the over-generalization issue.

Table 14: Results of multiple metrics (%) on the TSB-AD-U dataset. AR and AP denote AUC-ROC and AUC-PR, respectively. R-AR and R-AP represent R-AUC-ROC and R-AUC-PR, respectively. V-ROC and V-PR correspond to VUS-ROC and VUS-PR, respectively.

| Metric | AnomalyTrans. | DCdetector | COUTA | MEMTO | D3R | DMamba | TimesNet | TFMAE | CATCH | Ours |
|---|---|---|---|---|---|---|---|---|---|---|
| AR | 42.89 | 48.25 | 47.36 | 50.22 | 58.36 | 49.66 | 55.89 | 48.99 | 65.90 | **67.50** |
| AP | 19.73 | 20.77 | 22.95 | 21.49 | 31.74 | 22.41 | 23.21 | 21.07 | 39.36 | **42.80** |
| F1 | 95.31 | 94.92 | 87.84 | 83.38 | 92.21 | 90.08 | 94.52 | 93.89 | 93.33 | **97.84** |
| R-AR | 49.79 | 50.58 | 46.88 | 53.84 | 58.72 | 50.01 | 54.40 | 53.60 | 72.63 | **73.11** |
| R-AP | 25.70 | 26.33 | 22.10 | 31.75 | 34.79 | 25.11 | 28.51 | 27.88 | 47.94 | **52.49** |
| V-ROC | 47.66 | 46.89 | 49.19 | 52.65 | 56.62 | 45.93 | 51.79 | 52.19 | **71.60** | 71.44 |
| V-PR | 23.01 | 21.57 | 23.55 | 26.19 | 30.41 | 20.29 | 24.88 | 25.59 | 44.06 | **48.58** |

Table 15: Results of label-based metrics (%) on five real-world datasets.

| Dataset | SMD | | | MSL | | | SMAP | | | SWaT | | | PSM | | |
|---|---|---|---|---|---|---|---|---|---|---|---|---|---|---|---|
| Metric | P | R | F1 | P | R | F1 | P | R | F1 | P | R | F1 | P | R | F1 |
| Isolation Forest | 42.31 | 73.29 | 53.64 | 53.94 | 86.54 | 66.45 | 52.39 | 59.07 | 55.53 | 49.29 | 44.95 | 47.02 | 76.09 | 92.45 | 83.48 |
| LOF | 56.34 | 39.86 | 46.68 | 47.72 | 85.25 | 61.18 | 58.93 | 56.33 | 57.60 | 72.15 | 65.43 | 68.62 | 57.89 | 90.49 | 70.61 |
| OC-SVM | 44.34 | 76.72 | 56.19 | 59.78 | 86.87 | 70.82 | 53.85 | 59.07 | 56.34 | 45.39 | 49.22 | 47.23 | 62.75 | 80.89 | 70.67 |
| AnomalyTrans. | 88.88 | 91.32 | 90.08 | 90.84 | 84.43 | 92.52 | 94.53 | 93.76 | 94.14 | 90.07 | 99.77 | 94.78 | 97.22 | 94.91 | 96.05 |
| DCdetector | 86.01 | 84.52 | 85.26 | 92.10 | 92.19 | 92.14 | 93.56 | 97.94 | 95.70 | 95.31 | 97.54 | 96.45 | 97.19 | 97.00 | 97.03 |
| COUTA | 75.83 | 75.83 | 77.46 | 91.09 | 90.75 | 90.92 | 80.56 | 74.01 | 77.15 | 95.36 | 68.79 | 81.51 | **99.76** | 86.77 | 92.81 |
| MEMTO | 87.96 | 96.58 | 92.47 | 91.00 | **95.28** | 93.04 | 93.84 | **99.65** | 96.66 | 92.33 | 99.08 | 95.91 | 97.41 | 98.10 | 97.75 |
| D3R | 87.74 | 96.09 | 91.91 | 91.77 | 94.33 | 93.03 | 92.23 | 96.11 | 94.21 | 83.09 | 83.00 | 83.04 | 93.84 | 99.11 | 96.45 |
| DMamba | 92.57 | 54.04 | 68.24 | **93.69** | 64.06 | 76.09 | **95.10** | 52.98 | 68.05 | 94.11 | 86.75 | 90.28 | 98.26 | 82.89 | 89.91 |
| TimesNet | **94.53** | 82.56 | 88.14 | 90.00 | 90.12 | 90.06 | 80.99 | 69.10 | 74.58 | 90.63 | 91.37 | 90.94 | 96.79 | 96.95 | 96.82 |
| TFMAE | 90.83 | 90.27 | 90.46 | 93.01 | 95.21 | **94.33** | 95.00 | 97.97 | 96.46 | 95.10 | 100.00 | 97.49 | 97.70 | 98.24 | 97.97 |
| CATCH | 45.38 | **98.22** | 62.08 | 66.86 | 94.28 | 78.24 | 82.42 | 56.88 | 67.31 | 84.94 | 90.39 | 87.58 | 96.68 | 98.23 | 97.45 |
| MFRM | 93.38 | 96.24 | **95.05** | 93.26 | 93.18 | 93.22 | 94.91 | 99.63 | **97.21** | 97.35 | 99.23 | **98.28** | 98.37 | 98.99 | **98.68** |

Table 16: Results of score-based metrics (%) on five real-world datasets. AR and AP denote AUC-ROC and AUC-PR.

| Dataset | SMD | | MSL | | SMAP | | SWaT | | PSM | | Average | |
|---|---|---|---|---|---|---|---|---|---|---|---|---|
| Metric | AR | AP | AR | AP | AR | AP | AR | AP | AR | AP | AR | AP |
| Isolation Forest | 66.38 | 12.18 | 52.39 | 11.38 | 48.67 | 12.20 | 34.61 | 9.26 | 54.16 | 33.41 | 51.24 | 15.69 |
| LOF | 54.87 | 8.29 | 48.77 | 10.63 | 42.10 | 10.87 | 48.82 | 11.33 | 51.25 | 30.05 | 49.16 | 14.23 |
| OC-SVM | 60.22 | 10.35 | 52.41 | 13.31 | 39.30 | 10.16 | 65.73 | 16.95 | 61.88 | 41.83 | 55.91 | 18.52 |
| AnomalyTrans. | 44.12 | 3.99 | 48.47 | 10.50 | 52.63 | 13.41 | 38.07 | 11.32 | 49.22 | 27.83 | 46.50 | 13.41 |
| DCdetector | 49.13 | 4.13 | 49.19 | 10.82 | **56.81** | **14.75** | 50.91 | 12.26 | 49.88 | 25.00 | 51.18 | 13.39 |
| COUTA | 49.70 | 4.14 | 50.25 | 10.55 | 49.53 | 12.77 | 49.43 | 12.13 | 51.78 | 30.29 | 50.14 | 13.98 |
| MEMTO | 50.12 | 4.86 | 49.84 | 10.57 | 50.92 | 12.95 | 77.46 | 33.72 | 50.25 | 28.15 | 55.72 | 18.05 |
| D3R | 64.20 | 12.24 | 65.26 | **16.99** | 41.35 | 10.62 | 56.65 | 13.30 | 50.03 | 26.31 | 55.50 | 15.89 |
| DMamba | 52.06 | 6.39 | 50.65 | 10.82 | 49.97 | 12.79 | 50.15 | 12.17 | 50.30 | 28.04 | 50.63 | 14.04 |
| TimesNet | 51.60 | 7.33 | 52.03 | 12.77 | 50.88 | 14.03 | 51.13 | 13.67 | 53.28 | 32.66 | 51.78 | 16.09 |
| TFMAE | 35.76 | 3.68 | 51.93 | 11.06 | 54.96 | 14.56 | 50.22 | 12.13 | 48.21 | 27.28 | 48.22 | 13.74 |
| CATCH | 78.74 | **16.35** | **66.21** | 14.53 | 53.11 | 14.37 | 39.19 | 22.66 | 65.83 | 43.96 | 60.62 | 22.37 |
| MFRM | **79.88** | 16.25 | 60.56 | 14.79 | 56.32 | 13.77 | **88.49** | **76.86** | **71.55** | **51.09** | **71.36** | **34.55** |

## C.12 COMPARISON OF THE PERFORMANCE AND EFFICIENCY OF DIFFERENT METHODS

For a comprehensive evaluation of various baselines and MFRM, we visualize in Figure 14 their F1 scores, inference time, and memory usage across five datasets. Overall, MFRM achieves optimal performance with comparatively lower inference time and reduced memory overhead.

Table 17: Results of multi metrics on two challenging datasets.

| Dataset | NIPS-TS-G. | | | NIPS-TS-S. | | |
|---|---|---|---|---|---|---|
| Metric | AR | AP | F1 | AR | AP | F1 |
| Isolation Forest | 61.87 | 5.18 | 39.06 | 50.03 | 31.25 | 58.34 |
| LOF | 63.37 | 4.32 | 38.55 | 45.77 | 30.92 | 55.90 |
| OC-SVM | 80.41 | 3.91 | 29.62 | 54.15 | 38.56 | 48.47 |
| AnomalyTrans. | 11.86 | 0.88 | 19.20 | 41.55 | 29.63 | 73.09 |
| DCdetactor | 40.30 | 0.95 | 28.35 | 42.63 | 29.89 | 73.56 |
| COUTA | 49.50 | 1.05 | 30.08 | 51.53 | 34.67 | 61.35 |
| MEMTO | 58.20 | 2.45 | 69.60 | 50.68 | 33.44 | 73.91 |
| D3R | 80.32 | 12.39 | 58.66 | 53.40 | 40.97 | 67.57 |
| DMamba | 63.23 | 14.85 | 44.23 | 51.41 | 34.44 | 74.53 |
| TimesNet | 55.33 | 10.37 | 25.38 | 54.49 | 41.31 | 65.81 |
| TFMAE | 49.29 | 1.07 | 42.87 | 48.35 | 32.18 | 73.43 |
| CATCH | **97.22** | **49.20** | 79.48 | 52.03 | 44.46 | **75.85** |
| MFRM | 87.18 | 31.67 | **80.18** | **70.77** | **61.93** | 74.79 |

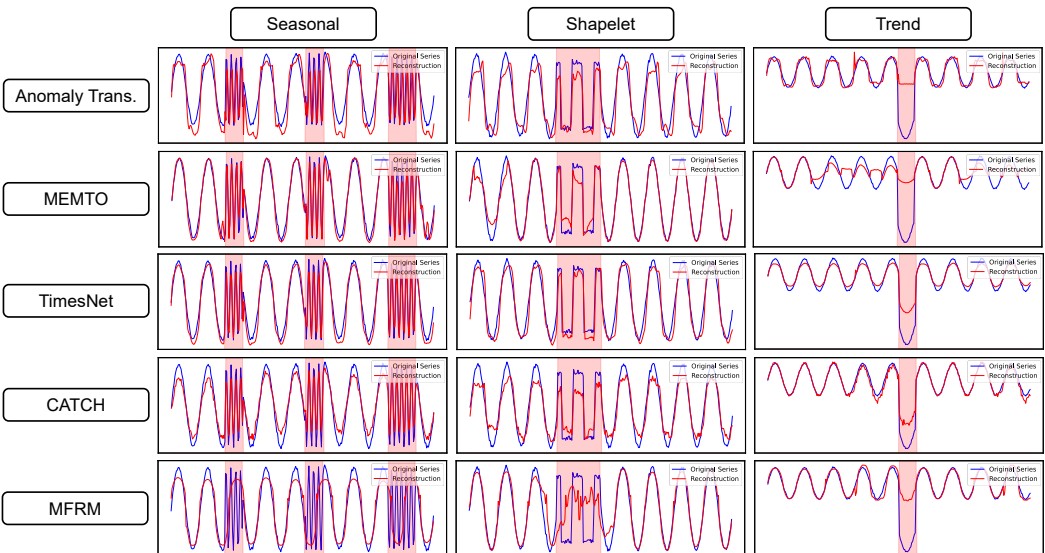

Figure 13: Comparison of the reconstruction results of three types of synthetic anomalies by five methods.

### C.13 VISUALIZATION OF ANOMALY DETECTION CAPABILITIES AT DIFFERENT STAGES

To analyze the contribution of each stage in MFRM to anomaly detection results, Figure 15 visualizes the performance of PTM, FRM, and the combined two-stage approach in detecting different types of anomalies.

As can be observed, the PTM stage demonstrates capability in detecting point anomalies but performs inadequately when confronted with the other three types of pattern anomalies (seasonal, shapelet, and trend), exhibiting notable false negatives and false positives. By leveraging fine-grained frequency information, the FRM stage proves more effective in detecting these three pattern anomalies, though it still retains a certain degree of false positives. The integrated two-stage approach ultimately yields precise and smooth anomaly scores, achieving simultaneous accurate detection of both point anomalies and complex pattern anomalies.

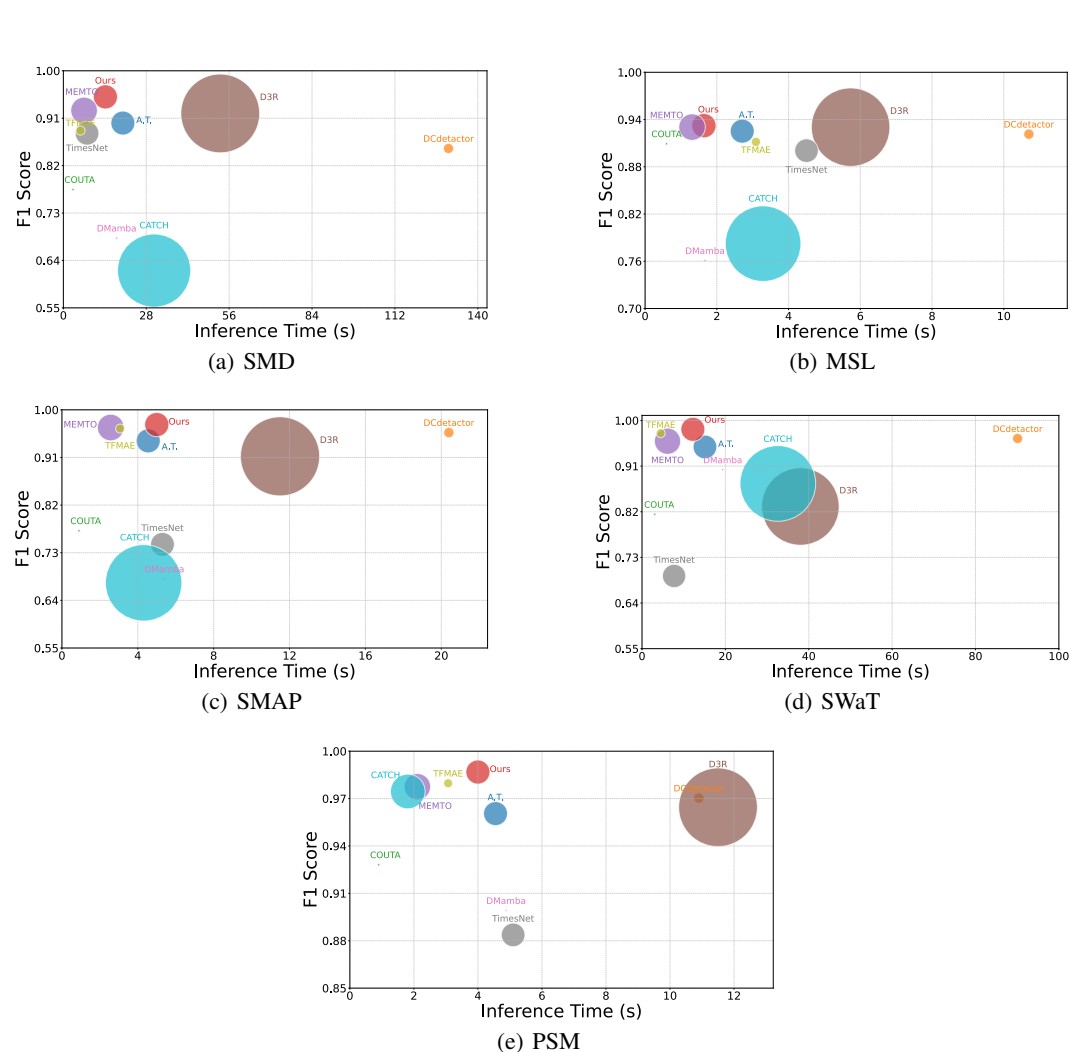

Figure 14: Comparison of the performance and efficiency of different methods on five datasets. The larger the circle, the greater the memory usage.

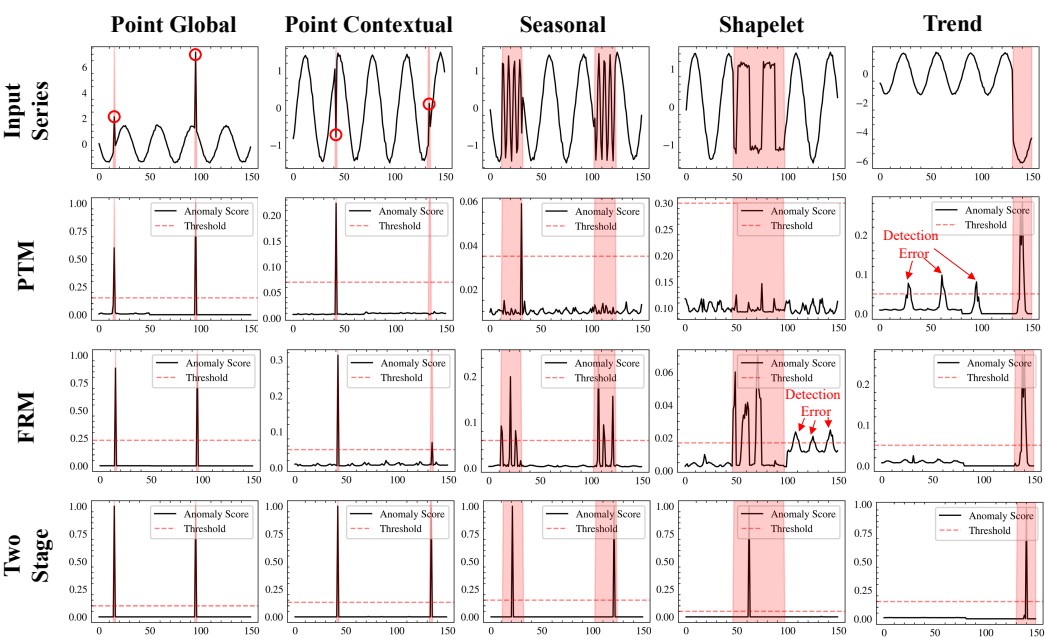

Figure 15: The effects of PTM, FRM and the two-stage combination in detecting different types of anomalies.

