# OpenReview forum: "MFRM: Masked Frequency-Refined Modeling for Multivariate Time Series Anomaly Detection"
_ICLR.cc/2026/Conference — Submitted to ICLR 2026_

### Official Review · Reviewer_yqLW · 2025-10-24

**Soundness:** 3
**Presentation:** 3
**Contribution:** 2
**Rating:** 6
**Confidence:** 4

**Summary:**

This paper focuses on reconstruction-based time series anomaly detection in the frequency domain, where not all frequency components contribute equally, and over-generalized reconstruction models may also reconstruct anomalies well. To address these issues, it proposes a self-learning model generating a mask in the frequency domain, and demonstrates its effectiveness on several widely used anomaly detection benchmarks.

**Strengths:**

1. The paper is logically clear, well-structured, and easy to follow. In addition, the figures are well-designed and visually appealing.

2. The proposed model is technically sound, with intuitive technical rationality and reliability.

3. The paper shows certain advantages over state-of-the-art methods. Moreover, by comparing the proposed mask generation approach with other alternatives, the authors demonstrate its effectiveness.

**Weaknesses:**

1. In the introduction, the authors aim to address two issues — that not all frequency components are equally important and that the model tends to over-generalize. However, the experiments do not directly validate whether these two problems have been effectively solved.

2. The motivation for addressing these issues is not sufficiently elaborated. For example, regarding the unequal importance of frequency components, the paper does not clearly explain the negative effects this problem may cause; for the over-generalization issue, it remains unclear whether current SOTA methods indeed suffer from this problem, which could have been supported by empirical evidence.

3. In the methodology section, the paper primarily focuses on how the proposed method works, but rarely discusses why the proposed design can effectively solve the stated problems.

**Questions:**

1. Could the authors provide additional experiments to demonstrate that the baseline methods indeed exhibit over-generalization, leading to anomalies being well reconstructed?

2. Could the unequal importance of frequency components be discussed in more detail, particularly regarding its negative impact on time series anomaly detection?

3. Could the authors show the time overhead and memory overhead of MFRM and baselines?

---

> ### Author Response · Authors · 2025-11-17
> **Response to Reviewer yqLW (Part 1)**
>
> Thank you for your careful review and valuable feedback!
>
> **General Response:**
> 1. **Restatement of issue 1:** It should be clarified that the statement "not all frequency components are equally important" describes the background of the problem rather than the problem itself. **The actual issue** lies in the fact that unimportant or redundant frequencies can interfere with the model's granularity of frequency perception, particularly for models that operate across the full frequency spectrum [1]  [2]. Since anomalies in time series often manifest as abnormal behaviors only within specific frequency bands, directly utilizing the full frequency range can impair the sensitivity of anomaly frequency detection.
>
> 2. **The focus of MFRM:** The primary focus of this paper is on extracting and leveraging fine-grained frequency information. "Importance" serves as the criterion for extracting such fine-grained frequencies — we aim to retain those frequencies that are highly correlated with normal patterns. At the same time, "unimportant" or redundant frequencies **are not inherently harmful to anomaly detection**; rather, they divert the model's "attention," and it is the **lack of granularity** that ultimately impacts anomaly detection performance.
>
> [1] Timesnet: Temporal 2d-variation modeling for general time series analysis. ICLR 2023
>
> [2] CATCH: Channel-aware multivariate time series anomaly detection via frequency patching. ICLR 2025
>
> **W1:** The experiments do not directly validate whether these two problems have been effectively solved.
>
> Following the restatement of **General Response** to the issue 1, we have specifically discussed in **Figures 5 and 10** how MFRM achieves fine-grained utilization of frequency information. Additionally, in **Appendix C.6**, we explain how this fine-grained approach benefits anomaly detection. As for the issue of over-generalization, we demonstrate in **Figure 4** that MFRM fundamentally addresses this problem through frequency masking.
>
> **W2:** The motivation for addressing these issues is not sufficiently elaborated.
>
> 1. **Issue 1:** We would like to reiterate the response to issue 1: the core focus of this work lies in the fine-grained utilization of frequency information, rather than establishing a strict dichotomy of importance among frequency components. However, providing a detailed discussion on the negative effects of coarse-grained frequency usage in anomaly detection remains hard, as it is difficult to address through theoretical analysis and lacks intuitive manifestation. The most direct validation comes from the performance comparison between MFRM and other frequency-based methods like CATCH, TimesNet and TFMAE in **Tables 1–3**, as well as the visualization in **Figure 8**, which compares how MFRM and TFMAE detect different types of anomalies.
>
> 2. **Issue 2:** Thank you for this valuable suggestion. We agree that a concrete discussion of whether SOTA methods truly suffer from over-generalization is necessary to clarify the motivation and depth of our work. Accordingly, we have added a comparative analysis in **Figure 13 (Appendix C.11) of the revised version**, which visually demonstrates the over-generalization issue in several reconstruction-based SOTA methods and highlights the advantage of MFRM in mitigating this problem.
>
> **W3:** In the methodology section, the paper primarily focuses on how the proposed method works, but rarely discusses why the proposed design can effectively solve the stated problems.
>
> In the Method section, we primarily elaborate on the workflow and design objectives of MFRM. As for the effectiveness of the design, we provide confirmation in experiments.
>
> **The learnable frequency masking module** plays a key role in addressing the two proposed issues by enabling adaptive frequency extraction and filtering. In **Figure 9 (Appendix C.5) and Figure 10 (Appendix C.6)**, we further explain how this frequency masking mechanism resolves these challenges and contributes to improving anomaly detection performance.
>
>  **The two-stage architecture** of MFRM is designed to refine the traditional temporal reconstruction models through fine-grained frequency utilization, while the incorporation of **adversarial learning** aims to strengthen the robustness of the framework. Their effectiveness are verified through the ablation experiments provided in **Table 4**.

---

> ### Author Response · Authors · 2025-11-17
> **Response to Reviewer yqLW (Part 2)**
>
> **Q1:** Could the authors provide additional experiments to demonstrate that the baseline methods indeed exhibit over-generalization, leading to anomalies being well reconstructed?
>
> Please refer to our response in **W2.2**.
>
> **Q2:** Could the unequal importance of frequency components be discussed in more detail, particularly regarding its negative impact on time series anomaly detection?
>
> Please refer to the **General Response** and response in **W2.1**.  However, it may be further clarified that the "unimportant" redundant frequencies essentially represent components with **low relevance to the description of normal patterns**.  These frequencies inherently carry limited informational value.
>
> Meanwhile, certain anomaly-related frequencies that are uncorrelated with normal patterns may also be categorized as "unimportant" redundant components during inference, as shown in **Figure 9 and Figure 11**.  As a result, filtering these frequencies helps enhance the distinction of anomalies.  Therefore, studying the positive impact of ignoring "unimportant" frequencies on anomaly detection holds greater significance than analyzing their potential negative effects.
>
> **Q3:** Could the authors show the time overhead and memory overhead of MFRM and baselines?
>
> Thank you for your valuable suggestion, which is crucial for a comprehensive analysis of our MFRM's performance. In response, we have included a new comparison in the **revised version** **(Figure 14)**, which evaluates the F1 score, inference time, and memory usage of MFRM against other baseline methods. Overall, the results demonstrate that MFRM achieves the best performance with lower inference time and reduced memory overhead.

---

> ### Comment · Reviewer_yqLW · 2025-11-26
> **Thank you for your response**
>
> Thank you for your response. However, my original concern has not been fully addressed, and your explanation raises additional questions that further deepen my concerns about the paper.
>
> W1: Figure 5 only demonstrates that the proposed method is able to extract different frequencies across different datasets. However, this does not provide sufficient evidence that the specific frequency range extracted for a given dataset is indeed the most effective or optimal for the downstream task. In other words, the figure shows variability, but not effectiveness.
>
> Moreover, Figure 10 lacks sufficient clarity for proper interpretation. First, it is unclear what exactly each scatter point represents. Second, the meanings of the horizontal and vertical axes are not clearly defined. More importantly, under the proposed learnable masking strategy, the distributions of masked-normal and masked-anomaly appear to be largely overlapping. If this is the case, it is questionable whether such a representation can effectively discriminate positive and negative samples in practice.
>
> W2 & Q1 & Q2
>
> Issue 1:
> According to the authors’ response, the potential negative impact of using coarse-grained frequency information in anomaly detection can be neither theoretically verified nor intuitively demonstrated through experiments. Consequently, the actual existence of this issue currently lacks both theoretical justification and clear empirical evidence. This makes it difficult to assess whether the problem is fundamentally valid or practically significant.
>
> Issue 2:
> The authors’ response to this issue deal with my concern.
>
> W3
>
> The authors’ response to this issue deal with my concern.
>
> Q3
>
> The authors’ response to this question deal with my concern.

---

> ### Author Response · Authors · 2025-11-26
> **Response to Reviewer yqLW (W1)**
>
> Thank you for your comment. We will further address your remaining concerns regarding the paper.
>
> **W1：**
> First, we would like to offer a detailed explanation of **Figure 10**. **Figure 10** presents t-SNE visualizations [1] of both normal and anomalous samples in the dataset before and after frequency masking, where each point represents an individual sample. t-SNE is a dimensionality reduction technique, which we employed to project each high-dimensional sample into a two-dimensional space for visualization purposes. The horizontal and vertical axes in the plot correspond to the two-dimensional reduced representations of the data and do not carry specific physical meanings.
>
> Second, since the MFRM is a **reconstruction-based** model rather than a classification-based one, the significant overlap between the distributions of masked normal and masked anomalies is **key to its success and actually benefits anomaly detection**. After frequency masking, the distribution of anomalous samples shifts toward the normal distribution, resulting in overlap, while significantly deviating from their original anomalous distribution (as shown by the purple and orange distributions in **Figure 10**). This makes it difficult for the MFRM to subsequently reconstruct the original anomalous samples based on such near-normal distributions, thereby allowing anomalies to be more easily detected through reconstruction error.
>
> Moreover, **Figure 10** demonstrates the effectiveness of fine-grained frequency extraction from two perspectives:
>
> 1. Applying frequency masking to normal samples results in high-density representations, indicating that the extracted frequency information can effectively concentrate and represent normal patterns in the data.
>
> 2. Applying frequency masking to anomalous samples disrupts their original anomalous distributions, causing them to shift closer to the normal distribution. This facilitates easier identification of anomalies via reconstruction error.
>
> As you pointed out, **Figure 5** illustrates variability rather than effectiveness.  Our previous response—`we have specifically discussed in Figures 5 and 10 how MFRM achieves fine-grained utilization of frequency information` —is intended to convey that **Figure 5** demonstrates, through the shown variability, MFRM’s capability for fine-grained utilization of frequency information, while **Figure 10** further reveals the specific characteristics of the extracted fine-grained frequency information and validates its effectiveness for anomaly detection.
>
> [1] Visualizing data using t-sne. Journal of Machine Learning Research 2008

---

> ### Author Response · Authors · 2025-11-26
> **Response to Reviewer yqLW (W2)**
>
> **W2 Issue1：**
> Thank you for your detailed comment! Our previous response might not have been complete enough. We will provide a more detailed explanation here.
>
> Indeed, previous studies [1] [2] [3] did not consider the granularity of frequency information utilization, which represents a methodological shortcoming. The impact of this limitation on anomaly detection often manifests as suboptimal detection performance. As we described in the paper:
>
> > their frequency modeling also lacks sufficient granularity, limiting their ability to detect complex anomalies through frequency details.
>
> Coarse-grained approaches can lead to certain **anomalies being undetected** or **normal instances being falsely detected** (as shown in the results of TFMAE in **Figure 8**). In our previous response, we regarded coarse-grained methods as merely "suboptimal" rather than causing clearly "negative" impacts detrimental to the task. Therefore, we previously provided the following response:
>
> > However, providing a detailed discussion on the negative effects of coarse-grained frequency usage in anomaly detection remains hard, as it is difficult to address through theoretical analysis and lacks intuitive manifestation.
>
> However, the paper ultimately demonstrates that coarse-grained methods are ”suboptimal“, and shows that MFRM achieves improved anomaly detection through fine-grained frequency utilization:
>
> 1. The performance of models utilizing coarse-grained frequency information, such as TimesNet [1], TFMAE [2], and CATCH [3], as shown in **Tables 1, 2, and 3**, are weaker than that of MFRM.
>
> 2. The comparison in **Figure 8** between TFMAE and MFRM in detecting different types of anomalies. TFMAE has abnormal missed detections and normal misjudgments, whereas MFRM demonstrates more accurate detection.
>
> In summary, we believe that the description of Issue 1 in the paper is accurate—that is, **coarse-grained frequency utilization can limit the model’s anomaly detection capability**. This is supported by the numerical results presented in **Tables 1, 2, and 3**, as well as the anomaly detection discrepancies illustrated in **Figure 8**, which confirm both the existence of the issue and the effectiveness of MFRM in addressing it. It is worth noting that our analysis approaches the problem from the perspective of improving “suboptimal” anomaly detection performance, rather than overcoming a “negative” impact.
>
> [1] Timesnet: Temporal2d-variationmodelingforgeneral timeseriesanalysis. ICLR 2023
>
> [2] Temporal-Frequency Masked Autoencoders for Time Series Anomaly Detection. ICDE 2024
>
> [3] CATCH: Channel-aware multivariate time series anomaly detection via frequency patching. ICLR 2025

---

### Official Review · Reviewer_3Cxo · 2025-10-31

**Soundness:** 2
**Presentation:** 2
**Contribution:** 2
**Rating:** 2
**Confidence:** 4

**Summary:**

The paper proposes a novel reconstruction model that strategically leverages frequency-domain information to enhance anomaly detection. It introduces a learnable frequency masking module to extract frequency components most correlated with normal behavioral patterns, thereby enabling the utilization of fine-grained frequency details.

**Strengths:**

S1. A learnable frequency masking module is proposed to fully exploit the discriminative capability of finegrained
frequency details.

S2. By disrupting the original spectrum through the frequency masking mechanism, the reconstruction
difficulty for anomalies in the time domain is exacerbated, thereby alleviating the over-generalization problem
of anomalies.

**Weaknesses:**

W1. What specific frequency components are considered redundant, and how does such redundancy affect
anomaly detection performance?

W2: The notation in Section 3 is inconsistent. The paper defines the x_i, but later (e.g., Eq. (5)) uses X_t,:. These
expressions are semantically equivalent, yet the inconsistent use of indices and subscripts introduces
unnecessary confusion.

W3. In Section 3.1, the Primary Temporal Modeling (PTM) stage employs only a simple embedding layer
followed by a standard Transformer for sequence reconstruction.

W4. The paper states that the proposed method is inspired by MCM, which was shown to be effective for
tabular anomaly detection. However, MCM’s validation is based on structured tabular data, which differ
significantly from time series in both distributional and anomalous characteristics. Why MCM can be directly
transferred to time series anomaly detection and is equally applicable and advantageous?

W5. How is the set of learnable vectors R generated?

W6. The experimental evaluation includes comparisons with nine deep learning–based methods but omits
classical statistical learning baselines (e.g., Isolation Forest, LOF, OCSVM). These methods are still
representative in early MTSAD research and sometimes outperform deep models in certain regimes.

**Questions:**

Q1: Which specific frequency ranges or feature patterns are referred to as redundant frequency information,
and how are these redundant components identified or determined?

Q2: Why do the authors use inconsistent mathematical notation throughout the paper?

Q3: Could the authors further elaborate on the design motivation of the PTM stage and clarify its independent
role within the overall model framework?

Q4: Why are classical statistical learning methods not included in the comparison?

---

> ### Author Response · Authors · 2025-11-16
> **Response to Reviewer 3Cxo (Part 1)**
>
> Thank you for your careful review and thoughtful feedback, which has been instrumental in helping us improve the quality of our paper.
>
> **W1:** What specific frequency components are considered redundant, and how does such redundancy affect anomaly detection performance?
>
> **Definition and identification**：
> 1.  In previous studies [1], redundant frequencies are simply defined as components with low magnitude in the amplitude spectrum, which can be identified using straightforward Fourier transform combined with thresholding.
>
> 2.  Redundant frequencies in this paper further refer to those components that can be removed without significantly affecting the quality of normal series reconstruction or the core informational content. As shown in **Figure 5**, after training on normal data, the proposed learnable frequency masking module can automatically extract key frequencies for modeling normal patterns , frequencies with low extraction probabilities can be considered redundant frequency components.
>
> **Affect**：
> 1.  For models [2]  [3]  [4] that perform frequency-domain modeling across the full frequency band, due to the lack of fine granularity, such redundant frequencies can interfere with their perception of frequency bands that significantly characterize anomalies, thereby undermining the objective of leveraging frequency details for anomaly detection. As illustrated in **Figure 8**, TFMAE [4] exhibits certain limitations, including false positives under normal conditions and missed detections in abnormal scenarios.
>
> 2.  As for MFRM, the filtered redundant frequencies yield higher reconstruction error on normal series compared to extracted frequencies (**Table 13**).
>
> [1] FITS: Modeling time series with $10k$ parameters. ICLR 2024
>
> [2] Timesnet: Temporal 2d-variation modeling for general time series analysis. ICLR 2023
>
> [3] CATCH: Channel-aware multivariate time series anomaly detection via frequency patching. ICLR 2025
>
> [4] Temporal-Frequency Masked Autoencoders for Time Series Anomaly Detection. ICDE 2024
>
> **W2:** The notation in Section 3 is inconsistent.
>
> We sincerely apologize for the inconsistent notations in the previous version of Section 3. While $x_i$ and $X_{t,:}​$ are semantically equivalent, the inconsistency indeed introduced unnecessary confusion. In the **revised version**, we have addressed this issue by unifying all notations to the new representation $x_t$​. Thank you once again for your careful review.
>
> **W3:** The Primary Temporal Modeling (PTM) stage employs only a simple embedding layer followed by a standard Transformer for sequence reconstruction.
>
> The core design of MFRM lies in its learnable frequency masking module rather than constructing complex model architectures.  Despite its structurally simple foundation, PTM possesses well-founded design motivations and demonstrated practical effectiveness.
>
> **(a) Clarification**: As described in the caption of **Figure 2**, the MFRM consists of only one Transformer encoder, where both the PTM and FRM **share the same Transformer.**  This point has been emphasized in the **revised Section 3** to avoid potential misunderstanding among readers:
> >
> > As previously mentioned, MFRM is a reconstruction model that employs a learnable frequency masking module as a bridge to refine normal series modeling via a two-stage architecture, where both stages share the same transformer.
>
> **(b) Design motivation**: The design motivation of PTM is to enable the transformer to preliminarily learn normal patterns through a "pre-training" style reconstruction in the time domain, thereby focusing primarily on dominant low-frequency information.  On this basis, the FRM refines the PTM by incorporating fine-grained frequency details extracted through frequency masking, thereby enriching the frequency representation.
>
> **(c) Independent role**: Within the overall MFRM framework, the PTM serves as a **foundational step** that paves the way for subsequent refined modeling.  Although the structure of PTM is simple, it helps to clarify the logic of the two-stage design in MFRM—i.e. , refining time-domain modeling with fine-grained frequency information.
>
> **(d) Effectiveness of PTM**: Ablation studies in **Table 4** demonstrate that using FRM alone without PTM leads to inferior performance.  This also indicates that the inclusion of PTM is practically effective—not only does it complete the logical flow of refined modeling, but it also contributes tangibly to the model's performance.

---

> ### Author Response · Authors · 2025-11-16
> **Response to Reviewer 3Cxo (Part 2)**
>
> **W4:** Why MCM can be directly transferred to time series anomaly detection and is equally applicable and advantageous?
>
> Indeed, our work is inspired by the concept of "learnable masking" in MCM, but instead of directly transferring its complete framework to time series (e.g., by masking multivariate time series variables), we introduce a novel design tailored to the specific challenges of time series anomaly detection. The core differences between the two are as follows.
>
> | Aspect         | MCM                               | MFRM                                  |
> | -------------- | --------------------------------- | ------------------------------------- |
> | Masking Target | Tabular Features                  | Time Series Frequency bands           |
> | Masked Type    | Correlated features               | Redundant frequencies                 |
> | Learning Goal  | Identify informative correlations | Extraction and filtering of frequency |
>
> **(a) Mechanism differences**: Due to the orthogonality of frequency components [1], masked modeling in the frequency domain does not focus on correlations between frequencies. Instead, by reconstructing the original series from the frequency-masked ones, the model learns to focus on frequencies that are highly relevant to normal patterns, as illustrated in **Figure 10**.
>
> **(b) Design motivation**: Certain complex temporal anomalies manifest as frequency patterns that deviate from normal behavior [2].  Through frequency masking, such anomalous frequency components are filtered out, making it harder to reconstruct anomalous series and thereby enhancing the discriminative capability of the model.
>
> **(c) Validity verification**: The efficacy of MFRM has been extensively validated through a series of experiments, as demonstrated in **Figures 4, 9, 10, and 11**.
>
> The effectiveness of MFRM stems from its consideration of the unique characteristics of time series anomaly detection, rather than a direct adoption of the MCM approach.
>
> [1] Revisiting Attention for Multivariate Time Series Forecasting. AAAI 2025
>
> [2] CATCH: Channel-aware multivariate time series anomaly detection via frequency patching. ICLR 2025
>
> **W5:** How is the set of learnable vectors R generated?
>
> The learnable vector R is essentially a parameter matrix initialized using Kaiming initialization, and does not involve specific generative logic.
>
> **W6:**  The experimental evaluation includes comparisons with nine deep learning–based methods but omits classical statistical learning baselines (e.g., Isolation Forest, LOF, OCSVM).
>
> Thank you for your suggestion.  Indeed, in several recent works such as AnomalyTransformer [1], DCdetector [2], MEMTO [3], and D3R [4], classical statistical learning methods appear to underperform, which is the reason we do not initially include them in our comparison.  However, after carefully considering your feedback, we observe that these studies primarily use the F1-score for evaluation, which may provide an incomplete perspective.
>
> Therefore, in the **revised version**, we supplement our experimental results by including three classical methods—Isolation Forest, LOF, and OC-SVM—across seven datasets in **Tables 15, 16, and 17**.  These methods are now evaluated using not only the F1-score but also AUC-ROC and AUC-PR for a more comprehensive comparison.
>
> The results indicate that although classical methods generally achieve lower F1-scores compared to deep learning approaches, they demonstrate competitive performance in terms of AUC-ROC and AUC-PR, particularly on the SMD and PSM datasets.  We sincerely appreciate your valuable suggestion, which significantly improves the quality and completeness of our work.
>
> [1] Anomaly Transformer: Time Series Anomaly Detection with Association Discrepancy. ICLR 2022
>
> [2] DCdetector: Dual Attention Contrastive Representation Learning for Time Series Anomaly Detection. KDD 2023
>
> [3] MEMTO: Memory-guided Transformer for Multivariate Time Series Anomaly Detection. NIPS 2023
>
> [4] Drift doesn't Matter: Dynamic Decomposition with Dffusion Reconstruction for Unstable Multivariate Time Series Anomaly Detection. NIPS 2023

---

> ### Author Response · Authors · 2025-11-16
> **Response to Reviewer 3Cxo (Part 3)**
>
> **Q1:** Which specific frequency ranges or feature patterns are referred to as redundant frequency information, and how are these redundant components identified or determined?
>
> Please refer to **W1**'s reply.
>
> **Q2:** Why do the authors use inconsistent mathematical notation throughout the paper?
>
> Please refer to **W2**'s reply.
>
> **Q3:** Could the authors further elaborate on the design motivation of the PTM stage and clarify its independent role within the overall model framework?
>
> Please refer to **W3**'s reply.
>
> **Q4:** Why are classical statistical learning methods not included in the comparison?
>
> Please refer to **W6**'s reply.

---

> > ### Comment · Reviewer_3Cxo · 2025-11-27
> >
> > I have read the author rebuttal. Thank you for the clarification.
> >
> > However, it is not clear why the proposed frequency masking method can ensure that the selected frequencies are all those related tonormal conditions, while also filtering out frequencies that deviate from normal. The paper does not clearly explain this point.
> >
> > Where the novelty of the proposed learnable frequency masking? How is it fundamentally different from existing frequency-domain masking methods? The apparent difference from CATCH seems to bethe use of learnable vectors R, yet R involved does not incorporate any special design.
> >
> > How is the proportion of the loss function? As shown in Table 4, the impact of different modules on performance varies significantly. More qualitative case studies will help clarify how each of the twostage reconstructions contributes to the detection results.
> >
> > In Table 1, P, R, and F1 seem to have adopted the point adjustment strategy, but this strategy has been clearly demonstrated in previous works to lead to incorrect performance evaluations. Even if only one point in an abnormal segment is correctly monitored, point adjustment assumes that the model has correctly monitored the entire segment for abnormalities, which is highly unreasonable.

---

> > > ### Author Response · Authors · 2025-11-28
> > > **Kindly Request for Reviewer's Feedback**
> > >
> > > Dear Reviewer 3Cxo,
> > >
> > > As the rebuttal period is drawing to a close, we want to kindly follow up and check whether our response has adequately addressed your main concerns.   If so, we would be very grateful if you could consider adjusting your score accordingly.    Should you have any further suggestions or points for discussion, we are more than willing to continue improving the paper and make any additional revisions.
> > >
> > > Thank you once again for the time and thoughtful feedback you have devoted to our work during this busy period!
> > >
> > > Warm regards, Authors

---

> ### Author Response · Authors · 2025-11-27
> **Response to Reviewer 3Cxo**
>
> Thank you for your further comments.
>
> ## ①
> The key reason why the learnable frequency masking can ensure that the selected frequencies are related to normal patterns while filtering out those deviating from normality lies in its **"learnable" nature and optimization on normal data**, as described in our paper:
>
> > Since the learnable frequency masking is optimized on normal series, it tends to select frequencies corresponding to normal behavior while filtering out “unseen” frequencies deviating from normal patterns.
>
> During training, to reconstruct the original series from the masked frequencies, the model learns to prioritize those key frequencies that highly represent normal patterns. As a result, during inference, it naturally gains the ability to filter out frequencies that deviate from normality.
>
> To empirically validate the effectiveness of this unique characteristic of the learnable frequency masking, we provide detailed discussions in **Appendix C.6 (Figure 10), Appendix C.5 (Figure 9), and Appendix C.7 (Figure 11)**.
>
> - **Figure 10** presents t-SNE visualizations of both normal and abnormal samples from the dataset, before and after frequency masking. It can be observed that after masking, both normal and abnormal samples form distributions with higher density (green and purple). Moreover, the distribution of abnormal samples shifts closer to the normal distribution and shows a clear deviation from the original abnormal distribution. This confirms that the frequencies selected by the masking are highly informative and strongly correlated with normal patterns.
>
> - **Figure 9** compares the original spectra of synthetic anomalies with the spectra after MFRM masking, while **Figure 11** demonstrates the ability of MFRM to detect complex anomalies caused by spectral differences. Together, the spectral comparisons before and after masking illustrate that the extracted frequencies are indeed relevant to normal conditions, while anomalous frequencies deviating from normality are effectively filtered out, thereby enhancing the discriminability of abnormal samples.
>
> ## ②
> **Novelty**: The proposed learnable frequency masking generates a learnable hard mask using binarized activation. Its generation process relies on a **learnable vector R and an attention mechanism**. The core of its novelty lies in its **"learnable"** nature and its design tailored for specific frequency spectra.
>
> **Differences**: Existing frequency masking methods are relatively scarce, with representative approaches including the Top-K masking used in TFMAE [1], the random masking employed in FEI [2], and the learnable masking adopted in CATCH [3].
>
> - Compared to TFMAE and FEI, the essential distinction of MFRM's approach similarly  lies in its **"learnable"** characteristic. **Table 5** quantitatively demonstrates the advantages of our learnable method over these two non-learnable methods. Additionally, **Figure 10** qualitatively shows the benefits of our learnable approach compared to the Top-K strategy.
>
> - While CATCH also employs a learnable masking strategy and is a model that operates in the frequency domain, please note that **our method is fundamentally different from it**:
>
>     1. The masking in CATCH is applied to the attention map, representing an improvement over the traditional causal mask in a learnable manner. Its attention map measures correlations between different patches in the series, and this process is conducted in a latent space, thereby losing the interpretability of the original frequency spectrum. Therefore, the mask used in CATCH is essentially a **"patch mask" rather than a true "frequency mask"**. In contrast, the mask in MFRM operates directly on the frequency spectrum and is a genuine frequency-aware mask, thus offering stronger frequency-domain interpretability for anomaly detection (as shown in **Figures 9 and 11**).
>
>     2. CATCH utilizes the Gumbel-Softmax technique to ensure gradient propagation, while MFRM employs a binarized activation function. **Table 5** also includes a comparison with the Gumbel-Softmax approach, demonstrating the superiority of our method.
> [1] Temporal-Frequency Masked Autoencoders for Time Series Anomaly Detection. ICDE 2024
>
> [2] Frequency-masked embedding inference: A non-contrastive approach for
> time series representation learning. AAAI 2025
>
> [3] CATCH: Channel-aware multivariate time series anomaly detection via frequency patching. ICLR 2025

---

> ### Author Response · Authors · 2025-11-27
> **Response to Reviewer 3Cxo**
>
> ## ③
> 1. As directly indicated in **Equation 11**, the three loss components are summed directly without any proportional weighting.
>
> 2. Thank you for your insightful consideration! In addition to quantitative analysis, it is valuable to clarify the contribution of each stage in MFRM to the final detection results through qualitative studies, which helps to intuitively understand the practical role of each component. **In Figure 15 of Appendix C.13** in the **revised version**, we visualize the anomaly detection results of the PTM, FRM, and their two-stage combination:
>     - The PTM is capable of detecting point anomalies but performs poorly on the other three types of pattern anomalies (seasonal, shapelet, and trend), with noticeable false negatives and false positives.
>
>     - The FRM utilizes fine-grained frequency information, making it more effective in detecting the three pattern anomalies, though it still exhibits a certain level of false alarms.
>
>     - The two-stage combined approach ultimately produces accurate and smooth anomaly scores, achieving precise detection of both point anomalies and complex pattern anomalies simultaneously.
>
> ## ④
> The rationale for applying point adjustment follows the description in Anomaly Transformer [1]:
>
> > This strategy is justified by the observation that an abnormal time point will trigger an alert and subsequently draw attention to the entire segment in real-world applications.
>
> However, this adjustment strategy also carries certain risks, as it may lead to misleading performance evaluations [2, 3]. We have also taken note of this issue, which is one of the main reasons why we additionally report AUC-ROC and AUC-PR metrics in **Table 2 and Table 3**. Furthermore, in **Appendix C.1 (Table 10)**, we provide a comprehensive evaluation of MFRM's performance using various other metrics, such as VUS [4]. As stated in the original text:
>
> > However, many studies have demonstrated that point adjustment can lead to biased performance evaluations. To ensure a fair assessment of MFRM, we employ multiple recently proposed metrics in Appendix C.1.
>
> In summary, this paper does not rely solely on the point-adjustment-based F1-score to evaluate model performance. Instead, the superior performance of MFRM is comprehensively demonstrated through a combination of **multiple evaluation metrics** in **Table 2, Table 3, and Table 10**.
>
> [1] Anomaly Transformer: Time Series Anomaly Detection with Association Discrepancy. ICLR 2022
>
> [2] Drift doesn’t matter: Dynamic decomposition with diffusion reconstruction for unstable multivariate time series anomaly detection. NIPS 2023
>
> [3] Local evaluation of time series anomaly detection algorithms. KDD 2022
>
> [4] Volume under the surface: a new accuracy evaluation measure for time-series anomaly
> detection. VLDB 2022

---

### Official Review · Reviewer_MKcd · 2025-10-31

**Soundness:** 3
**Presentation:** 3
**Contribution:** 2
**Rating:** 6
**Confidence:** 4

**Summary:**

This paper proposes MFRM (Masked Frequency-Refined Modeling), a reconstruction-based method for multivariate time series anomaly detection that addresses two key challenges: traditional models' bias toward low-frequency bands while underutilizing fine-grained frequency details, and the over-generalization issue where models reconstruct both normal and anomalous data well. The core innovation is a learnable frequency masking module that adaptively extracts frequency components correlated with normal patterns and filters out anomalous frequencies, combined with a two-stage architecture (PTM for initial reconstruction, FRM for frequency-refined modeling) trained with adversarial learning. The method disrupts the original spectrum of anomalous series through frequency masking, making their reconstruction more difficult in the time domain. Experiments on seven benchmark datasets demonstrate state-of-the-art performance, with average improvements of 10.74% in AUC-ROC and 12.18% in AUC-PR over the previous best method.

**Strengths:**

1. Well written
2. A lot of experiments and discussions
3. The hard frequency masking is novel

**Weaknesses:**

1. The frequency masking seems to mask high-frequency components in all the visualizations shown. This is obvious and similar to what a model in the time domain would do (although it's easier to do so in the frequency domain).
2. None of the experiments include statistical significance tests. For example, they do not report the results of 10 runs with mean and standard deviation to test for significance.
3. The performances are not that great.

**Questions:**

1. Why is the anomaly score multiplicative? With a multiplicative anomaly score, all three loss components have to be high to be classified as an anomaly. This doesn't make sense to me, since a time series might only have an anomaly in either the time domain or the frequency domain, but not both.
2. Are the Transformers in PTM and FRM shared?
3. There might be some anomalies in the training set for some of the datasets. How does that affect the behavior of frequency masking?

---

> ### Author Response · Authors · 2025-11-17
> **Response to Reviewer MKcd (Part 1)**
>
> Thank you for your careful review and thoughtful suggestions.
>
> **W1:** The frequency masking seems to mask high-frequency components in all the visualizations shown.
>
> Our visualization of the frequency masking is primarily presented in **Figures 5, 9, 11, and 12**.
>
> - As shown in **Figure 5 (Section 4.3.2)**, the frequency mask demonstrates varying degrees of attention to components across different frequency bands on all five datasets, without completely masking high-frequency components.
>
> - Results on synthetic anomalies in **Figure 9 (Appendix C.5)** demonstrate that the masks effectively preserve normal frequency patterns while filtering out unseen anomalous frequencies—this behavior is not a simple distinction between high and low frequencies, as masking also occurs in low-frequency components such as shapelet and trend.
>
> - The case study in **Figure 11 (Appendix C.7)** further illustrates the same point as Figure 9.
>
> - The spectral comparison in **Figure 12 (Appendix C.8)** highlights the difference between the spectrum of traditional temporal reconstruction (red curve), which mainly focuses on low-frequency regions, and the spectrum of MFRM’s frequency masking (blue curve), which adaptively selects frequencies across both low and high bands.
>
> Overall, frequency masking does not merely mask high-frequency components but extracts those highly relevant to the normal patterns while filtering out the rest.
>
> **W2:** None of the experiments include statistical significance tests.
>
> As reported in **Appendix B.4**, all experimental results of MFRM represent the average values over three runs. More importantly, since we fix all random seeds, the results across runs are nearly identical. Therefore, in line with the reporting practices adopted in references [1], [2], and [3], we report our results without including statistical significance tests.
>
> [1] Anomaly Transformer: Time Series Anomaly Detection with Association Discrepancy. ICLR 2022
>
> [2] MEMTO: Memory-guided Transformer for Multivariate Time Series Anomaly Detection. NIPS 2023
>
> [3] CATCH: Channel-aware multivariate time series anomaly detection via frequency patching. ICLR 2025
>
> **W3:** The performances are not that great.
>
> Although MFRM does not achieve the best results across all metrics on every dataset (as shown in **Tables 1, 2, 3, 10, and 14**), it remains highly competitive in terms of average performance across datasets — with average improvements of **10.74% in AUC-ROC and 12.18% in AUC-PR** over the previous best method. Moreover, MFRM also outperforms other approaches on novel evaluation metrics such as R-AR, R-AP, V-ROC, and V-PR, as demonstrated in **Tables 10 and 14**.

---

> ### Author Response · Authors · 2025-11-17
> **Response to Reviewer MKcd (Part 2)**
>
> **Q1:** Why is the anomaly score multiplicative? With a multiplicative anomaly score, all three loss components have to be high to be classified as an anomaly.
>
> As you rightly point out, combining the three loss components multiplicatively to form the anomaly score requires all three components to be relatively high for an anomaly to be detected — since an extremely low value in any component would suppress the final result.
>
> **(a) Design motivation**: However, as shown in **Figure 8 (Appendix C.3)** and supported by related works such as AnomalyTransformer [1], MEMTO [2], TFMAE [3], and CATCH [4], **misclassification of normal instances** often poses a more critical challenge in MTSAD than missing true anomalies. Therefore, the multiplicative aggregation scheme is intentionally designed to **reduce false alarms on normal samples**, thereby improving overall detection performance — especially in cases where relying on a single type of loss component may be insufficient.
>
> **(b) Validity verification**: As discussed in **Appendix C.3**, the MixScore effectively suppresses false detections while successfully identifying diverse anomaly types, thereby leading to improved anomaly detection performance.
>
> **(c) Further analysis**: We also recognize that the motivation behind the multiplicative anomaly score may not be entirely robust. To address this, we conduct ablation studies on MixScore in **Table 11 (Appendix C.4)**. The results confirm that multiplicatively aggregating the three loss components yields the best overall performance.
>
> [1] Anomaly Transformer: Time Series Anomaly Detection with Association Discrepancy. ICLR 2022
>
> [2] MEMTO: Memory-guided Transformer for Multivariate Time Series Anomaly Detection. NIPS 2023
>
> [3] Temporal-Frequency Masked Autoencoders for Time Series Anomaly Detection. ICDE 2024
>
> [4] CATCH:Channel-aware multivariate time series anomaly detection via frequency patching. ICLR 2025
>
>
> **Q2:** Are the Transformers in PTM and FRM shared?
>
> Yes, exactly! As described in the caption of **Figure 2**, MFRM is essentially a two-stage method implemented with a single Transformer encoder, where both the PTM and FRM share the same Transformer. The distinction between the two stages lies solely in their inputs—respectively utilizing self-attention and cross-attention. This design better highlights that MFRM builds upon traditional time-domain reconstruction (PTM) and refines it with frequency-domain details (FRM).
>
> Considering your question, we have further emphasized this point in the **Section 3** of the **revised version**:
> >
> >  As previously mentioned, MFRM is a reconstruction model that employs a learnable frequency masking module as a bridge to refine normal series modeling via a two-stage architecture, where both stages share the same transformer.
>
> Additionally, we conduct comparative experiments in **Appendix B.5 (Table 9)** and **Appendix C.4 (Table 12)**, evaluating both the original encoder-decoder structure (where PTM and FRM use separate Transformers) and the single-Transformer design of MFRM in terms of efficiency and performance. The results consistently demonstrate the superiority of the MFRM design.
>
> **Q3:** There might be some anomalies in the training set for some of the datasets. How does that affect the behavior of frequency masking?
>
> The potential interference of training set containing anomalies on the frequency masking process can be analyzed from two perspectives:
>
> First, the learnable frequency mask is optimized based on the average loss across all sequences. If the anomaly contamination rate remains within a small and acceptable range, its impact on the behavior of the frequency mask is limited. In extreme cases where all training data are anomalous, the frequency mask would naturally learn frequency patterns that represent anomalies.
>
> Second, the frequency mask takes the magnitude spectrum in the frequency domain as input. If the injected anomalies do not cause significant changes in the frequency spectrum — for example, in the case of adding white noise with limited fluctuation — their influence on the mask's behavior will also be minimal.
>
> In summary, as long as the anomaly contamination rate in the training set remains within an acceptable range and these anomalies do not cause notable spectral changes, the behavior of the frequency mask will be largely unaffected.

---

### Official Review · Reviewer_uh39 · 2025-11-10

**Soundness:** 3
**Presentation:** 3
**Contribution:** 3
**Rating:** 6
**Confidence:** 2

**Summary:**

The paper presents MFRM, a reconstruction-based framework for multivariate time series anomaly detection that leverages adaptive frequency masking. A learnable module selects task-relevant spectral components to counter over-generalization and low-frequency bias in prior models. The two-stage architecture consists of temporal reconstruction and frequency-refined modeling with adversarial attention alignment, The proposed method achieves state-of-the-art results on seven benchmarks.

**Strengths:**

•	Identifies and visualize the issues of low-frequency bias and reconstruction over-generalization.
•	The proposed frequency masking module with attentive routers and binarization is well-designed.
•	Strong performance across multiple datasets and metrics, also with ablation and analysis of sensitivity.

**Weaknesses:**

-	There is limited discussion for the stability of the the score aggregations used in MixScore. Could further include the analysis on alternatives, such as weighted sum.
-	There is no deep theoretical analysis of critical properties such as convergence under adversarial learning or guarantees regarding the selection of “meaningful” frequencies.
-	Figure 5 shows that frequencies selected by the module vary across datasets. However, there is limited insight into the resulting frequency masks correspond to interpretable spectral regions or others.

**Questions:**

1.	How does inference scale with time series length and dimensionality?
2.	If selecting alternative aggregation methods, how about the performance of MixScore?

---

> ### Author Response · Authors · 2025-11-17
> **Response to Reviewer uh39 (Part 1)**
>
> We sincerely appreciate your thorough review and valuable feedback.
>
> **W1:** There is limited discussion for the stability of the the score aggregations used in MixScore.
>
> As you rightly pointed out, the multiplicative aggregation method used in MixScore indeed requires further discussion to validate its stability. In **Appendix C.4 (Table 11)**, we provide a detailed analysis of the anomaly detection performance differences among the components of MixScore and various aggregation methods. The results indicate that while the summation-based approach does not significantly underperform compared to the multiplicative method—and even performs better on the MSL dataset—but we still recommend the multiplicative aggregation method when considering the average performance overall.
>
> **W2:** There is no deep theoretical analysis of critical properties such as convergence under adversarial learning or guarantees regarding the selection of “meaningful” frequencies.
>
> Indeed, providing a theoretical analysis of the convergence of adversarial loss and the selection of "meaningful" frequencies is relatively challenging. We approach these issues more from a experimental perspective.
>
> **(a) Adversarial learning**: The process of adversarial learning is essentially a minimax two-player game [1], where the final model reaches a competitive equilibrium state. However, it is difficult to provide a precise theoretical representation of this equilibrium (e.g., convergence to a specific distribution). In **Appendix C.2 (Figure 7)**, we illustrate the variation of adversarial loss during training across five datasets to facilitate an intuitive understanding. It can be confirmed that the adversarial loss of MFRM eventually oscillates within a very small range, which ensures the stability of both the training process and the final model.
>
> **(b) Meaningful frequencies**: Driven by the optimization objective of reconstructing the original sequence from frequency-masked ones, MFRM learns to focus on frequency components that are valuable for reconstructing normal sequences. In **Appendix C.6 (t-SNE Visualization, Figure 10)**, we observe that the extracted frequency components form a high-density representation that characterizes normal patterns. In **Appendix C.6 (Reconstruction Error Analysis, Table 13)**, we find that reconstructing with the selected frequencies yields errors smaller than those using "filtered" frequencies across all datasets. These experimental results empirically confirm that the selected frequencies are indeed "meaningful."
>
> [1] Generative Adversarial Nets. NIPS 2014
>
>
> **W3:** However, there is limited insight into the resulting frequency masks correspond to interpretable spectral regions or others.
>
> Regarding the interpretability of the frequency components selected by the frequency mask and those filtered out, we discuss this in **Figure 10 (Appendix C.6)** and **Figure 9 (Appendix C.5)**, respectively.
>
> First, from the t-SNE visualization of the selected frequency components (**Figure 10, right**), it can be observed that the frequency mask acts similarly to a gating mechanism, extracting and compressing high-density, important information. Driven by this mechanism, the reconstruction model's ability to distinguish between normal and abnormal samples is enhanced.
>
> Second, **Figure 9** illustrates how the spectrum of the "remaining regions" filtered out by the frequency mask contributes to anomaly detection. Since certain frequency components that may easily indicate anomalies often differ from normal frequency patterns, they tend to reside in these filtered "remaining regions." By filtering out such anomalous frequencies, the model finds it more difficult to accurately reconstruct abnormal sequences. This, in turn, improves anomaly detection performance by avoiding over-generalization.

---

> ### Author Response · Authors · 2025-11-17
> **Response to Reviewer uh39 (Part 2)**
>
> **Q1:** How does inference scale with time series length and dimensionality?
>
> **(a) Sequences of different lengths**: Since MFRM models sequences along the multivariate dimension, changing the sequence length is equivalent to changing the batch size for MFRM. This only alters the amount of data without affecting the size of the latent space. Therefore, the inference performance of the model exhibits nearly linear scaling.
>
> **(b) Different dimensionalities**: If the dimensionality of the input sequence itself changes (i.e., the number of variables varies), only the size of the embedding layer’s mapping matrix in MFRM is affected, with almost no impact on inference performance. However, if the dimensionality of the latent space changes, as discussed in **Table 9 (Appendix B.5)**, the effect on inference performance is nonlinear. In particular, when the $d_{model}$ increases significantly, the inference burden is further exacerbated.
>
> **Q2:** If selecting alternative aggregation methods, how about the performance of MixScore?
>
> Please refer to **W1**'s reply.

---

### Meta-Review · Area_Chair_Qm4r · 2026-01-02

**Summary:**

The reviews are mixed, with one clear reject (3Cxo) and three accept-but-borderline assessments (uh39, MKcd, yqLW). While the paper is generally regarded as well written and the empirical section is extensive, the core concerns are about conceptual/technical clarity and substantiation of the claimed mechanism and motivation. In particular, the skeptical reviews converge on the point that the paper does not convincingly establish (i) why the proposed learnable frequency masking reliably selects *normal-related* frequencies and filters abnormal ones (beyond post-hoc visualizations), and (ii) why the *coarse-grained* frequency modeling issue is a distinct, well-evidenced problem rather than simply an empirical observation that some baselines underperform. Even after rebuttal, Reviewer yqLW remained unconvinced that the figures (esp. the t-SNE) provide actionable evidence of effectiveness, and Reviewer 3Cxo continued to question the novelty relative to existing frequency-domain masking approaches (e.g., TFMAE/FEI/CATCH) and the lack of a crisp explanation of what is fundamentally new beyond a learnable hard mask with auxiliary parameters.

On the evaluation side, although the authors added additional metrics beyond point-adjusted F1 and included classical baselines and resource overhead comparisons, doubts remain about whether the performance improvements are robust and cleanly attributable. Reviewers raised concerns about the multiplicative anomaly scoring (potentially suppressing anomalies that manifest primarily in one domain), the lack of statistical significance reporting (three runs with *fixed* seeds), and limited insight into how the selected masks map to interpretable spectral regions across datasets.

**Reviewer Concerns:**

Several concerns were meaningfully addressed in the rebuttal/revision.

However, several key issues remain outstanding and continue to drive uncertainty about acceptance. The main outstanding concerns are about novelty, motivation, and theoretical justification. In particular, it remains unclear why the proposed learnable frequency masking reliably selects frequencies tied to normal behavior and filters anomalous ones, beyond post-hoc visualizations; Reviewer 3Cxo explicitly remained unconvinced on this point. Also, the paper does not yet make a compelling case that its masking mechanism is fundamentally different from or more principled than existing frequency-domain methods (e.g., TFMAE, FEI, CATCH), as the use of learnable vectors alone is not seen as a strong novelty claim. Finally, Reviewer yqLW continued to question whether coarse-grained frequency modeling is a well-defined and practically significant problem rather than an empirical observation of suboptimal baselines, noting that the motivation and evidence remain indirect.

**Reviewer Scores:**

Reviewers uh39 and MKcd would keep his/her current rating.

Reviewer 3Cxo would keep his/her current rating or increase his/her rating from 2 to 4, because the main concerns were not clearly addressed.

Reviewer yqLW would even lower his/her rating from 6 to 4, because he/she raised additional concerns from the authors' rebuttals.

---

### Decision · Program_Chairs · 2026-01-26

Reject